# Advanced modeling of gas chemistry and aerosol dynamics with SSH-aerosol v2.0

Karine Sartelet[1], Zhizhao Wang[1,2,3], Youngseob Kim[1], Victor Lannuque[2], and Florian Couvidat[2]

[1]CEREA, Ecole nationale des ponts et chaussées, EDF R & D, Institut Polytechnique de Paris, IPSL, 77455 Marne-la-vallée, France
[2]INERIS, 60550 Verneuil en Halatte, France
[3]Now at University of California, Riverside, CA, USA, and National Center for Atmospheric Research, CO, USA

**Correspondence:** Karine Sartelet (karine.sartelet@enpc.fr), Florian Couvidat (florian.couvidat@ineris.fr)

**Abstract.** SSH-aerosol is developed to represent the evolution of primary and secondary pollutants in the atmosphere by processes linked to gas-phase chemistry, aerosol dynamics (coagulation, condensation/evaporation and nucleation) and intra-particle reactions. The representation of process complexity can be adjusted based on the user's choices.

The model uses a sectional size distribution, and offers the capability to discretize chemical composition to account for the mixing state of particles. The algorithms are designed to represent the evolution of ultrafine particles: conservation of mass and number during numerical resolution, taking into account the Kelvin effect, the condensation dynamics of nonvolatile compounds, and nucleation. Different parameterizations are provided for nucleation: binary, ternary, heteromolecular and organic nucleation depending on the compounds involved.

For gas-phase chemistry, schemes of different complexities can be handled: from simple schemes to model ozone, oxidants and inorganic chemistry (e.g. CB05, RACM2, Melchior2), to more complex schemes, e.g. from the Master Chemical Mechanism (MCM). The complexity of the schemes used for secondary organic aerosol (SOA) formation may also be adjusted: from schemes built from chamber data to near-explicit schemes from MCM. SOA schemes reduced using the GENOA algorithm are also provided for several precursors (toluene, a sesquiterpene and three monoterpenes), together with their evaluation against chamber or flow-tube experiments. A wall-loss module has also been added for easier comparisons to chamber experiments.

Specific developments were made in version 2.0 to automatically link the chosen gas-phase mechanism to SOA formation by using the SMILES structure of organic compounds, allowing for the determination of their hydrophilic and hydrophobic properties and for the partitioning in both organic and aqueous phases. The gas/particle partitioning may also be represented with different complexities. For the organic phase, viscosity may be modelled, adapting the aerosol viscosity to its composition, and coupling organic and inorganic thermodynamics. The dynamic evolution of the partitioning may be computed explicitly or thermodynamic equilibrium may be assumed. Different options are also provided to simulate the chemistry of organic compounds inside the particles with different types of reactions: irreversible $1^{st}$ order reactions, bulk oligomerization, hydratation of aldehydes and reactions of organic compounds with inorganic ions.

The SSH-aerosol model may be installed with a docker for standalone use. It has also been coupled to several 3D models to represent gas and aerosol concentrations: from the local scale with computational fluid dynamic and street network models to the regional scale with chemistry-transport models.

## 1 Introduction

The simulation of three-dimensional atmospheric pollutant concentrations is influenced by numerous factors, each associated with potentially significant uncertainties, including emissions, meteorological conditions, transport processes, and deposition. The formation and evolution of aerosols involve many chemical compounds and several complex processes, which are often very simplified in 3D air-quality and climate models, or even not taken into account, despite the fact that some of these processes may be critical for accurately representing aerosol properties and concentrations. The creation of aerosol box models allows for the independent evaluation of the various processes involved in aerosol dynamics and atmospheric chemistry, free from the uncertainties associated with 3D modeling and the computational limitations it imposes. They can therefore be used to simulate both aerosol formation and evolution in enclosed environment (like atmospheric smog chambers or flow tubes) with a detailed representation of processes, or inside 3D air quality models with generally more simplified approaches. Furthermore, the creation of box models enables the coupling of the same aerosol model with different 3D models across various scales (e.g. in regional and urban air quality models, Maison et al. (2024a)), ensuring a consistent representation of aerosol chemistry and dynamics.

Challenges in the development of aerosol box models are numerous: representation of the mixing state (Riemer et al., 2019), gas-particle interactions (Pöschl et al., 2007), formation of secondary inorganic and organic aerosols and their properties, formation and growth of ultrafine particles (Lee et al., 2019). Consequently, the choice of the model type and the aerosol representation is guided by the need to adequately represent and resolve these distinct challenges. Several aerosol box models have been developed, and most represent aerosol dynamics using a sectional approach (SIREAM (Debry et al., 2007), SCRAM (Zhu et al., 2015), ATRAS2 (Matsui, 2017), MAFOR (Karl et al., 2022), MOZAIC, SALSA (Kokkola et al., 2018), ARCA (Clusius et al., 2022), a modal approach (MAM (Sartelet et al., 2006), M7 (Vignati et al., 2004)) or a particle-based approach (Curtis et al., 2024).

In SSH-aerosol, the sectional approach was selected over the modal approach because it is numerically simpler and offers greater precision in representing the growth of ultrafine particles (Devilliers et al., 2013). The SCRAM module for aerosol dynamics is used because of its ability to represent the particle mixing state by discretizing both size and composition. Few models account for the mixing state; one example is ATRAS2, which specifically represents the mixing state of black carbon, and M7, which uses strong simplifying assumptions when mixing different modes. In SSH-aerosol, the mixing state of each compound can be represented explicitly, as in the SCRAM module, but modeling the mixing state may lead to longer computation times than in the classical case where aerosols are assumed to be internally mixed (Zhu et al., 2016).

For atmospheric chemistry, various mechanisms are commonly used to represent the formation of ozone and oxidants, using different strategies to lump volatile organic compounds (VOCs). Examples include CB05 (Yarwood et al., 2005), RACM2

(Goliff et al., 2013), SAPRC (Carter, 2010) and MELCHIOR2 (Derognat et al., 2003). For the formation of organic oxidation products that could be semi-volatile and partition to form secondary organic aerosols (SOAs), different gaseous oxidation mechanisms of varying complexity exist. Near-explicit mechanisms, such as the Master Chemical Mechanism (MCM) (Jenkin et al., 2012) and those generated by the Generator for Explicit Chemistry and Kinetics of Organics in the Atmosphere (GECKO-A) (Aumont et al., 2005; Camredon et al., 2007), involve a large number of compounds and reactions, whereas empirical mechanisms built from chamber experiments use only a limited number of compounds and reactions, e.g. the Volatile Basis Set VBS (Donahue et al., 2006), the 2-products (Odum et al., 1996) and surrogate (Couvidat et al., 2012) approaches. Recently, the development of the GENerator of reduced Organic Aerosol mechanisms (GENOA) has allowed the reduction of near-explicit scheme while retaining major formation pathways and key properties of the resulting secondary organic aerosol, such as functional groups (Wang et al., 2022, 2023b). Reduced mechanisms are built by reducing and lumping reactions and species using different strategies (e.g. lumping, replacing, jumping, removing) under a variety of environmental conditions. These reductions preserve the main chemical pathways and represent radical reactions, which may strongly impact the formation of organic particles (Wang et al., 2024). Conserving molecular properties of organic compounds allows the estimation of their partitioning in both aqueous and organic phases, the influence of non-ideality and organic-inorganic interactions (Kim et al., 2011). The partitioning between gas and particles may be strongly influenced by viscosity (Kim et al., 2019; Schervish et al., 2024) and intra-particle reactions (Pun and Seigneur, 2007), which are still very rarely taken into account in 3D models. In this paper, the term SOA is used in a broad sense to refer to the compounds involved in the formation of organic particles, regardless of whether they are in the gas or particle phase.

The SSH-aerosol model was built to address these different challenges, with the ability to adjust the model's complexity depending on the user's choice. Numerous test cases are provided to check the model accuracy and to illustrate its capacities. Section 2 summarizes the features from the previous version and describes the code structure as well as new features. Section 3 focuses on the use and evaluation of chemical schemes of different complexity for SOA formation. Section 4 describes the advance gas/particle partitioning taking into account viscosity and particle-phase reactions, and section 5 addresses the modeling of ultrafine particles. Tools for coupling to 3D models and applications using SSH-aerosol coupled to 3D models are listed in section 6.

## 2 Model structure and new features

As illustrated in Fig. 1, the model was initially based on the merge of three state-of-the-art models:

- SCRAM: The Size-Composition Resolved Aerosol Model (Zhu et al., 2015), written in Fortran 90, which simulates the dynamics (coagulation, condensation/evaporation and nucleation), classifying particles by both composition and size with the possibility to resolve the mixing state of atmospheric particles.

- SOAP: The Secondary Organic Aerosol Processor (Couvidat and Sartelet, 2015), a thermodynamic model that computes the partitioning of organic compounds, taking into account hygroscopicity, absorption into the aqueous and organic phases of particles, non-ideality, viscosity and phase separation. it is written in C++.

– $H^2O$: The Hydrophilic/Hydrophobic Organics (Couvidat et al., 2012) mechanism, which uses a molecular surrogate approach to represent the myriad of formation of semivolatile organic compounds formed from the oxidation in the atmosphere of volatile organic compounds. Gas-phase chemistry is solved using algorithms written in Fortran 90.

The size distribution is represented using a sectional approach (Debry et al., 2007), in which the particle population is approximated by discrete size sections characterized by their mass, number, and mean diameter. Depending on user settings, the

section boundaries may either remain fixed or evolve in time, and the mean diameter may be constrained to the geometric midpoint of the section or allowed to vary within its bounds. Particles are assumed to be spherical, so that the mass $M$, number $N$, and mean diameter $d_m$ in each section satisfy $M = N\rho\pi/6d_m^3$, with $\rho$ the particle density, which can be fixed or vary with the particle composition.

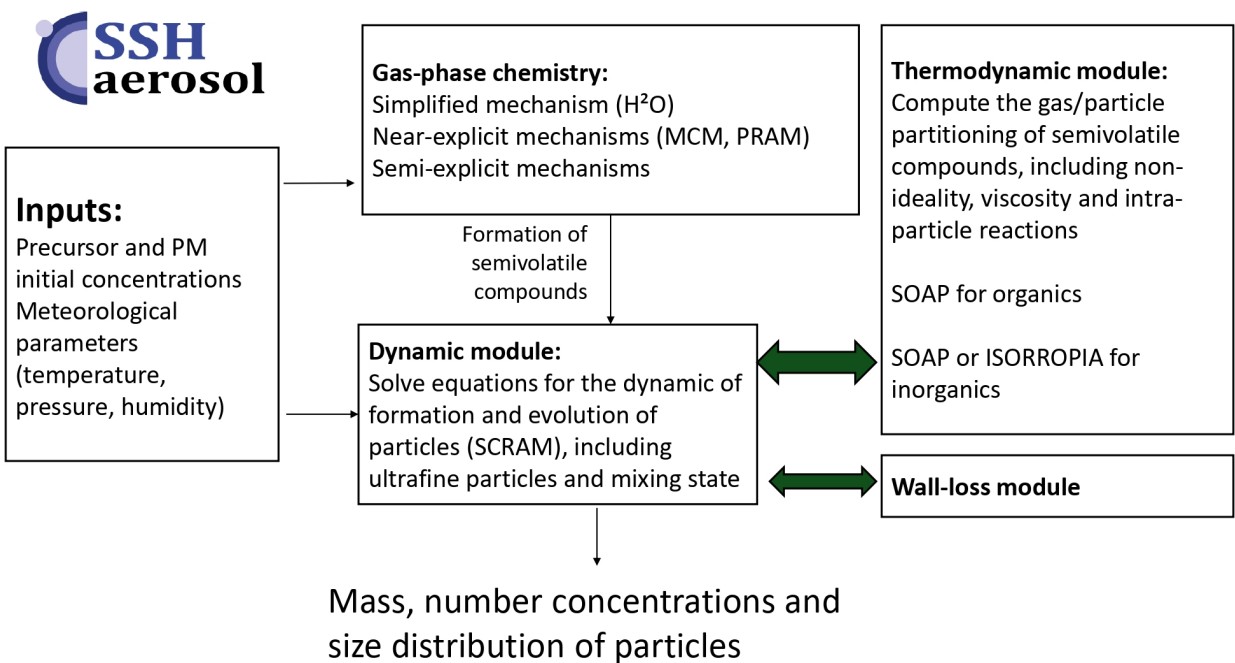

**Figure 1.** Workflow of the SSH-aerosol model.

## 2.1    Model structure

The SSH-aerosol package contains different directories for source code, configuration files, input files, output files, and output visualization. The model structure is the same as previous versions: the main program file is *ssh-aerosol.f90* where the configuration file is read, as well as the list of gaseous and particle compounds and their properties. The different variables, such as meteorological parameters and initial concentrations are read from files. Partition coefficients for coagulation (coefficients indicating how particles formed by the coagulation of two smaller particles should be redistributed into size sections) are com-

puted once at the beginning of simulations. They can be written into files, which can then be used to avoid their recalculation. The gaseous chemistry (e.g. $H^2O$ module) and aerosol dynamics (SCRAM and SOAP modules) are solved sequentially with a time step determined by the user. Note that this time step differs from the time steps used to solve gaseous chemistry and aerosol dynamics, which are automatically adapted, as detailed in section 2.3.

In SSH-aerosol version 1.1, gaseous chemical reactions were read by a pre-processor and interpreted before their use in SSH-aerosol. In version 2.0, this pre-processor is removed, and the chemical reactions are directly read and interpreted in SSH-aerosol, eliminating the need to recompile the program each time the chemical scheme changes.

Simulations require a *namelist* configuration file that details the chemical species and reactions, input and output conditions, as well as various simulation setups. The gas phase species are listed in the file *gas_phase_species* together with their molar mass, while aerosol species are detailed in a separate file *aerosol_species_list_file*. Each row represents an aerosol species and its specific properties: the species type (1 for dust, 2 for elemental carbon, 3 for inorganics and 4 for organics), the group to which the species belongs in case of mixing-state resolved particles, the name of corresponding gas-phase species, its molar weight (g/mol), the collision factor (probability that a particle–particle encounter leads to effective coalescence), molecular diameter (Angstrom), surface tension (N/m), accommodation coefficient (between 0 and 1), density in kg/m$^3$ and whether or not the species is nonvolatile. The column "partitioning" indicates whether the species condense on only an organic phase (keyword: HPHO), only an aqueous phase (keyword: HPHI) or both (keyword: BOTH). In the next column, the SMILES structure of the species or its decomposition into the 60 UNIFAC functional groups (see Table 4) can be given, then the saturation vapor pressure (in Torr), the enthalpy of vaporization, the Henry constant, and the reference temperature at which it is specified. Note that the Henry constant is not needed if both the saturation vapor pressure and the SMILES structure (or decomposition into functional groups) are provided. The saturation vapor pressures are provided as model inputs and may be derived through different methods or parameterizations, including tools such as UManSysProp (Topping et al., 2016).

As a new feature in SSH-aerosol v2.0, outputs can now be written in a NetCDF format rather than a text format. Writing NetCDF output files is faster than text or binary outputs. Furthermore, SSH-aerosol v2.0 can be installed using Docker, allowing for a consistent setup across different platforms (macOS, Windows, and Linux/Unix).

## 2.2 Physical processes

The processes included in SSH-aerosol, as well as the main options, are illustrated in Fig. 2. New processes and options are highlighted in red, while those that have been improved since version 1.1 are highlighted in orange. A wide variety of configuration files, i.e. "test cases", are presented in the guide (Sartelet et al., 2025a) to illustrate the model capabilities. These test cases are also used for model checks from one model version to the next. Only modifications that do not affect the reproduction of the test cases or that can be explained by model improvements or corrections are kept.

Several test cases were presented in Sartelet et al. (2020). They concern:

  – **Coagulation and condensation for internally-mixed particles or mixing-state resolved particles.** The test cases consider an initial size distribution of internally-mixed particles following the urban/hazy conditions of Seigneur et al. (1986);

Zhang et al. (1999), which represent stringent conditions for coagulation/condensation respectively. For the condensation test case, an initial concentration of sulfuric acid that condenses onto particles is specified (9.9 $\mu$g/m$^3$). The number and volume distribution after 12 hours of simulation are the same as in Zhang et al. (1999). These test cases are also extended to the case of particles made of two populations of different composition to illustrate how coagulation/condensation affects the mixing state of particles, with the number and mass distribution being a function of the size and fraction of the chemical compounds.

– *Condensation/evaporation: influence of redistribution.* Because of condensation/evaporation, the bounds of diameter sections may evolve (due to growth or shrinking of particles). Because they need to be fixed if both condensation/evaporation and coagulation are solved, as well as in 3D Eulerian models, redistributing the mass and number concentrations among fixed sections is necessary. This redistribution leads to numerical diffusion. Different algorithms for redistribution are proposed and illustrated using the condensation test case (the moving-diameter approach or the Euler-coupled approach).

– *Condensation/evaporation: thermodynamic equilibrium assumption versus dynamic.* To save computational time, gas-phase concentrations at the particle surface are often assumed to be at equilibrium with the bulk gas concentrations, simplifying the calculation of condensation/evaporation. Bulk particle equilibrium concentrations are distributed between the different sections, using weights that depend on the condensation/evaporation kernel of the condensation/evaporation rate. A hybrid method is used where particles above a certain cutoff diameter are treated dynamically, while smaller particles are assumed to be in thermodynamic equilibrium. A simulation of Tokyo's pollution episode of the June 25$^{th}$ 2001 (Sartelet et al., 2006) shows that while dynamic modeling is more accurate, it is slower compared to using thermodynamic equilibrium, with the hybrid method offering a compromise.

– *Condensation/evaporation: role of Kelvin effect.* To highlight the significance of the Kelvin effect in the growth of ultra-fine particles, a simulation is conducted based on the emission of particle of nonadecane (C$_{19}$H$_{40}$) from a diesel engine exhaust, described in Devilliers et al. (2013). Modeling condensation/evaporation dynamically, two simulations are performed: one including the Kelvin effect and one without. The results show that the Kelvin effect is crucial for accurately modeling the evolution of ultra-fine particles made of a semi-volatile compound.

– *Condensation/evaporation: influence of viscosity using a constant coefficient.* The mass transfer of hydrophobic organic aerosols can be significantly influenced by particle viscosity. To explore this, the condensation of organic surrogates with varying volatilities is studied for different viscosities, represented by varying diffusion coefficients (D). The particles are modeled with the approach of Couvidat and Sartelet (2015), which implicitly represents the diffusion of organic aerosol inside aerosol, with five aerosol layers. The results show that the condensation of low-volatility surrogates is unaffected by viscosity, while higher-viscosity particles (low D) may inhibit the condensation of more volatile surrogates.

– ***Condensation/evaporation: absorption into organic or aqueous phases, influence of ideality.*** The oxidation of VOCs produces less volatile compounds that may condense onto particles, increasing particle mass. To study this, experiments on exhaust emissions from a Euro 5 gasoline car are compared to chamber measurements of organic aerosol concentrations from Platt et al. (2013). The simulation shows that more than half of the particle mass after 5 hours comes from the condensation of aged intermediate/semivolatile VOCs (IVOCs/SVOCs). Additionally, the presence of ammonia ($NH_3$)

emissions significantly increased inorganic aerosol concentrations. The study also explored how the hygroscopicity of compounds affects their phase, showing that isoprene oxidation products prefer to condense onto aqueous phase. The impact of ideality on SOA formation shows that activity coefficients greatly influence the SVOC gas-to-particle partitioning, particularly in the aqueous phase.

– ***Coupling between nucleation, coagulation and condensation.*** SSH-aerosol model v1.1 incorporates homogeneous bi-

nary and ternary nucleation schemes. A test case is run under the hazy conditions of Zhang et al. (1999) solving simultaneously nucleation, condensation, and coagulation. The results show that ultra-fine particles grow over time because of the combined influence of the different processes. Furthermore, the growth is faster when considering extremely-low-volatility compounds, such as the Monomer surrogate (an ELVOC from terpene oxidation).

## 2.3  Numerical implementation

The gaseous chemistry and aerosol modules of SSH-aerosol are invoked in the routine ssh-aerosol.F90. Gaseous chemistry ($H^2O$ module) and aerosol dynamics (SCRAM and SOAP modules) are solved sequentially. The outer loop time step is user-defined and corresponds to the output frequency of concentrations; however, the internal time steps for gaseous chemistry and aerosol dynamics differ from this output interval. Each module selects its own adaptive time step based on its numerical requirements.

Because the governing equations are stiff, dedicated stiff solvers are employed. Gaseous and organic chemistry are integrated using the two-step solver of Verwer et al. (Verwer, 1994), which replaces the Rosenbrock scheme used in version 1. Aerosol dynamics are solved with the second-order explicit trapezoidal rule (ETR) (Ascher and Petzold, 1998; Sartelet et al., 2006). These algorithms ensure numerical stability and computational efficiency while conserving particle mass and number across the discrete size spectrum. For each process, the initial time step is set to the minimum value specified by the user, and it is

subsequently adjusted adaptively according to the user-defined numerical tolerance. In SCRAM, coagulation may be solved either simultaneously with nucleation and condensation/evaporation or treated separately, depending on the user's choice. When the processes are split, coagulation is computed first because it is the slowest of the three. Nucleation and condensation/evaporation, which compete for the same gas-phase precursors, are always solved simultaneously. When bulk thermodynamic equilibrium is assumed, the equilibrium calculation is performed at the end of each main time step, after resolving coagula-

tion, nucleation, and condensation/evaporation for the dynamic sections and the low-volatility species. Evaporation can lead to the complete depletion of certain size bins. To ensure numerical robustness in such cases, the model employs mass-threshold parameters (TINYM and TINYN) that control accuracy and prevent the accumulation of non-physical residual masses. For

layers inside particles, the volume ratio of layers is assumed to remain constant. Therefore, a layer cannot vanish. Depending on the user's choice, mass and number concentrations can be remapped onto the fixed size sections after each time step. This remapping/redistribution becomes mandatory when coagulation is activated, because the coagulation kernels and partition coefficients are precomputed for fixed bin boundaries.

## 2.4 New features

Some of the new features of v2.0 concern gas-phase chemistry for SOA formation, with the possibility to use near-explicit chemical schemes for SOA formation. The properties of the organic species can now be detailed in the configuration file using their SMILES (simplified molecular input line entry system) code or functional group distribution according to the UNIFAC (Unified Functional Activity Coefficient) nomenclature, as well as their saturation vapor pressure and enthalpy of vaporization. When provided, SMILES code are automatically decomposited in UNIFAC functional groups. Chemical schemes can be obtained from MCM, and various near-explicit schemes are added in the software, namely for the oxidation of toluene, naphtalene, monoterpenes and sesquiterpenes. More specifically, the new functionalities related to gaseous chemistry concern:

– The coupling to the GENerator of Reduced Organic Aerosol Mechanisms (GENOA) and Master Chemical Mechanism

– The treatment of specific MCM and GECKO-A kinetic rates

– The replacement of the ROS2 numerical solver for gaseous chemistry by the two-step solver (Verwer et al., 1996) to reduce computational time.

For easier comparisons with chamber experimental results, a wall-loss module for vapors and particles and numerous comparisons to chamber experiments have been added.

New features also concern the partitioning between gas and particle, more specifically

– Addition of an intra-particle reaction module (oligomerization, hydrolysis, and reactions between organic and inorganic ions).

– For organic thermodynamic, the particle viscosity can now be estimated depending on the particle composition

– Coupled inorganic-organic thermodynamic can be handled in SOAP.

– Addition of nucleation and numerical schemes to simulate ultrafine particles:

Other new features are related to the dynamics of particles and the modelling of ultrafine particles:

– Consideration of the Kelvin effect to determine size distribution of condensing and evaporating species when assuming thermodynamic equilibrium

– Faster computation of coagulation partition coefficients

– New parameterizations of nucleation

These new features are detailed and illustrated with test cases in the following sections.

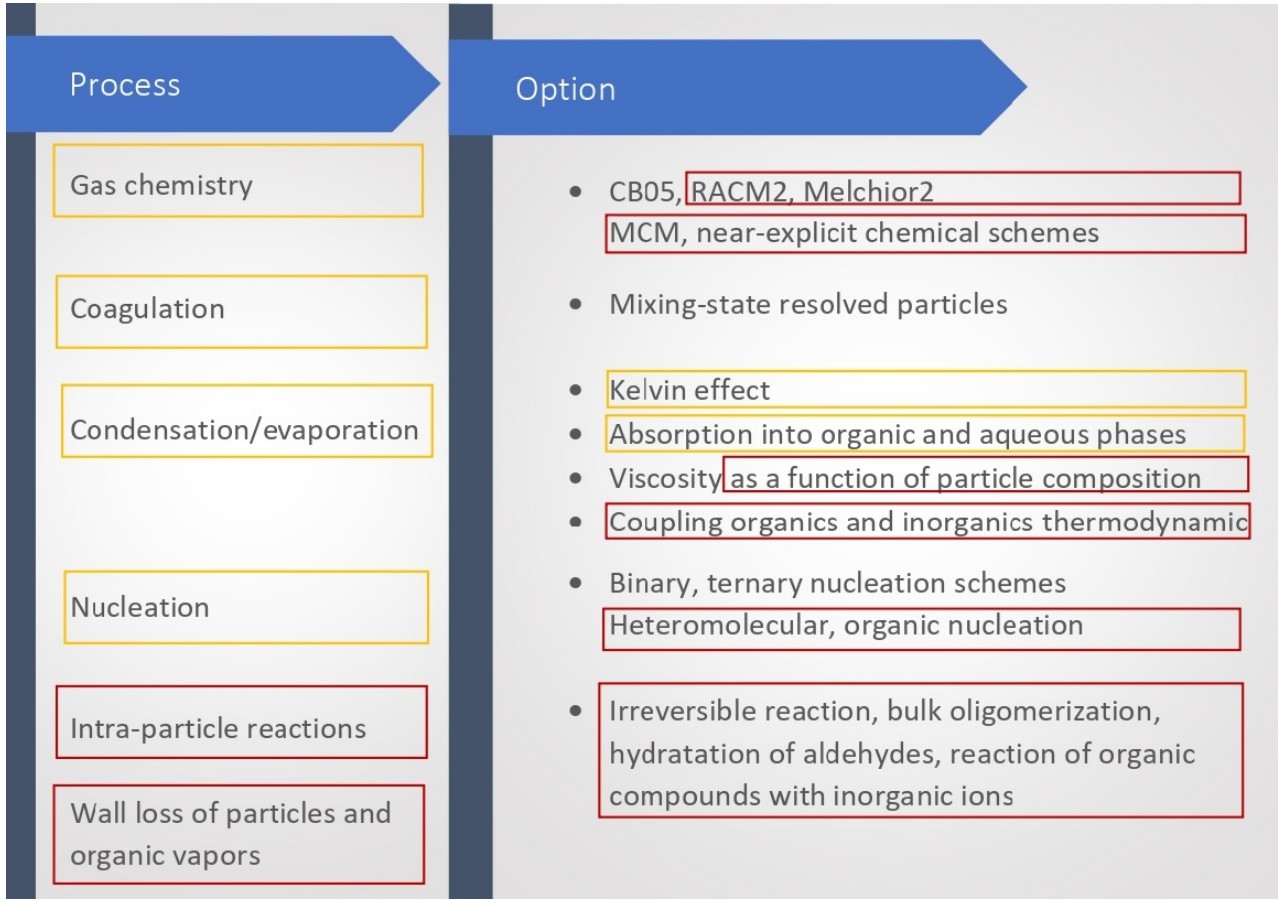

**Figure 2.** Main processes and options available in SSH-aerosol. New processes and options are highlighted in red, while those that have been improved since version 1.1 are highlighted in orange.

## 2.5   Runtime

As detailed in the appendix, many options are available, allowing the representation of the different physical processes with a large range of complexity. It is illustrated with different test cases in this paper. Increasing complexity may often lead to a large increase in computational time (CPU), as illustrated in Table 1. Near-explicit chemical schemes may be 3 to 65 times more expensive than parameterized schemes (comparisons of simulation numbers 1, 2,3; numbers 4 and 7). Taking into account ideality has a limited impact on CPU time, with an increase by a factor about 1.5-2 (comparisons of simulation numbers 4 and 5; numbers 4 and 6; numbers 9 and 10). The impact of oligomerisation on CPU time may be limited (comparison of simulation numbers 12 and 13), while the impact of viscosity may be high, with an increase by a factor almost 8 (comparison

of simulation numbers 10 and 11). Dynamic gas/particle partitioning may lead to a large increase of CPU time, from a factor 2 to a factor 20 (comparisons of simulation numbers 13 and 14; numbers 15 and 16; numbers 17 and 18; numbers 19 and 20). The relations between CPU time and model complexity presented here are only illustrative, and they may vary depending on the environmental conditions specified in the model's input.

**Table 1.** Runtime (in s) for different simulations presented in the Figures of this paper. Rdc. stands for reduced and Expl. for quasi-explicit.

| Simulation number | Simulation name | Figures | Runtime (s) |
|---|---|---|---|
| 1 | Rdc. SOA scheme | 4, 5 and 6 | 20 |
| 2 | Expl. SOA scheme | 4, 5 and 6 | 65 |
| 3 | $H^2O$ SOA scheme | 4, 5 and 6 | 1 |
| 4 | Expl. toluene SOA scheme - Ideal | 8a | 6 |
| 5 | Expl. toluene SOA scheme - Unifac | 8a | 10 |
| 6 | Expl. toluene SOA scheme - Aiomfac | 8a | 11 |
| 7 | Rdc. toluene SOA scheme - SART24-1 | 8b | 2 |
| 8 | Rdc. toluene SOA scheme - SART24-2 | 8b | 1 |
| 9 | Naphthalene SOA scheme - Ideal | 9 | 15 |
| 10 | Naphthalene SOA scheme - Aiomfac | 9 | 23 |
| 11 | Naphthalene SOA scheme - Aiomfac-visc | 9 | 162 |
| 12 | Isoprene SOA - Without oligomerization | 11 | 2 |
| 13 | Isoprene SOA - With oligomerization | 11 | 3 |
| 13 | $CaCO_3$ equilibrium no $HNO_3$ | 14 | 3 |
| 14 | $CaCO_3$ dynamic no $HNO_3$ | 14 | 28 |
| 15 | $CaCO_3$ equilibrium with $HNO_3$ | 14 | 2 |
| 16 | $CaCO_3$ dynamic with $HNO_3$ | 14 | 43 |
| 17 | Kelvin dynamic | 15 | 0.4 |
| 18 | Kelvin equilibrium | 15 | 0.2 |
| 19 | Kelvin dynamic | 16 | 2.7 |
| 20 | Kelvin equilibrium | 16 | 0.2 |

## 3  Chemical schemes for SOA formation

The oxidation of VOCs in the troposphere results in lower volatility organic compounds that can condense on particles under favorable conditions and form SOA. To address the influence of detailed gas-phase chemical processes of VOC oxidation, SOA formation may be modeled with schemes of different complexities. In SSH-aerosol v1.1, the $H^2O$ scheme (Couvidat et al., 2012; Sartelet et al., 2020) is available. This scheme was built from chamber experimental yields in order to represent

SOA formation with only a few chemical surrogate species. This scheme is suitable for 3D simulations because it allows a limited number of compounds to be used, which means that calculation time is affordable. This scheme also has other advantages, such as differentiation of low-NOx and high-NOx pathways, and the allocation of a molecular structure to most surrogate species with molecules, allowing to estimate the partitioning between gaseous and organic or gaseous and aqueous phases. However, the chemical pathways are too simplified, which makes it difficult to accurately predict future air quality, for example in the event of significant changes in NOx levels (Wang et al., 2024). In SSH-aerosol v2.0, semi-volatil organic compound formation can be represented using a range of chemical schemes, including highly detailed chemical mechanisms, such as those of the Master Chemical Mechanism (MCM) (Jenkin et al., 1997), as well as those presented in Lannuque et al. (2023); Wang et al. (2023b); Lannuque and Sartelet (2024), or schemes reduced from near-explicit chemical schemes using GENOA (Wang et al., 2023b; Sartelet et al., 2024). The SOA module may be customized to use more or less complex SOA chemical scheme depending on the SOA precursor.

The inorganic reactions governing oxidant formation do not always need to be included in the reaction list. Depending on the user's settings, oxidant concentrations can either be prescribed using constant input profiles, or they can be computed using existing gas-phase chemistry mechanism, such as CB05, RACM2 or MELCHIOR2. These schemes can be complemented with SOA chemical schemes describing the degradation of VOCs leading to SOA formation. To offer flexibility, the model allows users to construct customized chemical mechanisms, covering both gas-phase and SOA chemistry, by combining a chosen gas-phase mechanism with one or more SOA schemes (typically one per SOA precursor class). For VOCs already represented explicitly in the selected gas-phase mechanisms (e.g. toluene, monoterpenes), the associated SOA scheme can be added without modifying oxidant concentrations. However, some VOCs are not included in these gas-phase mechanisms, in which case the inclusion of specific reactions for radicals, oxidants as well as SOA formation may be necessary. For example, the impact of naphthalene on ozone and radical production is not represented in the CB05, RACM2 or MELCHIOR2 mechanisms.

Instructions for downloading the MCM chemical schemes from the MCM website and a converter for preparing input files in the SSH-aerosol format are detailed in the SSH-aerosol guide (Sartelet et al., 2025a). Since aerosol properties are not provided in MCM chemical schemes, users also need to define variables such as the method used to compute saturation vapor pressures for SVOCs. The UManSysProp (Topping et al., 2016) is used by the converter to compute aerosol-related properties accordingly, often using the vapor pressure estimation methods ('v0' (Myrdal and Yalkowsky, 1997)), with the boiling point estimation methods('b0' (Nannoolal et al., 2004)).

In the following section, a simulation with the MCM beta-caryophyllene scheme with fixed oxidant concentrations is first detailed. Then more general test cases illustrate the influence of the SOA scheme complexity on SOA formation depending on NOx levels.

## 3.1 Example of use of the MCM chemical scheme: beta-caryophyllene

The simulation of the beta-caryophyllene oxidation scheme (BCARY) in MCM v3.3.1 Jenkin et al. (2012) is illustrated by a test case. Users may choose to fix oxidant concentrations, in order to reduce SOA chemical schemes of a specific VOC using the GENOA tool (Wang et al., 2022, 2023b). For that application, the presence of $RO_2$ from other sources is taken into account

by introducing the notion of background $RO_2$. Indeed, as MCM reactions may involve radical reactions, SOA may be formed as a result of $RO_2$ species reacting with hydroperoxy radicals ($HO_2$) and other $RO_2$ radicals. Hence, $RO_2$-$RO_2$ reactions can be treated using the concept of a "$RO_2$ pool" that include $RO_2$ from all precursors. For example, a $RO_2$-$RO_2$ reaction can be written as follows:

```
NBCO2 -> 3.000E-01 BCBNO3 + 7.000E-01 BCAL + 7.000E-01 NO2
KINETIC RO2 1 ARR 9.200E-14
```

where ARR indicates it is an Arrhenius type of reaction, "RO2 1" indicates that the NBCO2 species reacts with the total concentration of all $RO_2$ species in the $RO_2$ pool with index "1". The number following "RO2" is required and specifies which $RO_2$ pool the species reacts with. It can be adjusted to refer to a different $RO_2$ pool if multiple pools are considered (the numbering starts from 1). All $RO_2$ pool species are defined in an input RO2 species file, where each line lists an RO2 species name and its corresponding $RO_2$ pool number, separated by a space.

The $RO_2$-$RO_2$ reaction can be treated in four different ways by adopting different options (*tag_RO2*) in the namelist:

- tag_RO2=0, compute no $RO_2$-$RO_2$ reaction.

- tag_RO2=1, compute $RO_2$-$RO_2$ reactions with only the produced $RO_2$ concentrations. The total concentrations of all $RO_2$ species recorded in the $RO_2$ species list are added up to compute the kinetic rates.

- tag_RO2=2, compute $RO_2$-$RO_2$ reactions with only the background $RO_2$ concentrations. For this option to work, the background concentrations should be provided for the gas-phase species *RO2pool* in the input constant concentration file.

- tag_RO2=3, compute $RO_2$-$RO_2$ reactions with both the produced and background $RO_2$ concentrations. In this case, the kinetic rate is calculated based on a sum of the background concentrations from the constant concentration file and the produced $RO_2$ concentrations of all $RO_2$ species.

In order to record the output concentrations of the $RO_2$ pool, a specific gas-phase species, *RO2pool*, has been added to the gas-phase species list.

The oxidation of BCARY for different *tag_RO2* options is illustrated in Fig. 3. The influence of the $RO_2$ pool option is low for BCARY, because the $RO_2$ + $RO_2$ pathways, i.e., the part influenced by the background $RO_2$ pool, is not the dominant pathway for BCARY SOA formation.

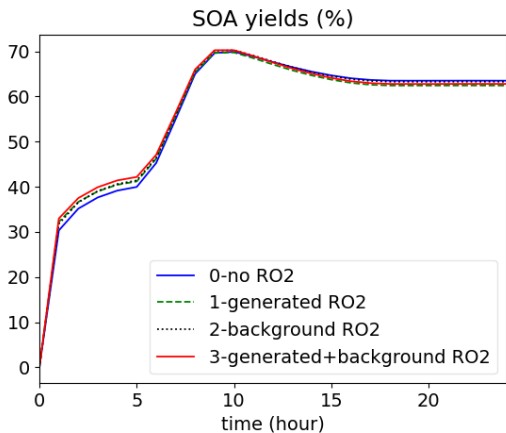

**Figure 3.** Time evolution of BCARY-SOA yields (the fraction of th BCARY's reacted mass that is converted into SOA) simulated with the BCARY MCM chemical mechanism and different $RO_2$-$RO_2$ reaction options.

### 3.2 Influence of SOA scheme complexity on SOA formation for different NOx levels

In these test cases, chemical schemes of different complexities are used and compared with respect to their simulated SOA concentrations for different $NO_2$ levels. Initial gas concentrations are taken to be representative of atmospheric conditions simulated by Sartelet et al. (2022) over Greater Paris in July 2009. The initial $NO_2$ concentration is taken equal to 26 $\mu$g m$^{-3}$,

and the simulated OH concentration varies between 1.2 and 7.6 x 10$^{-4}$ $\mu$g m$^{-3}$. Simulations are performed with initial $NO_2$ concentrations multiplied by 2 (52 $\mu$g m$^{-3}$) or divided by 2 (13 $\mu$g m$^{-3}$). Low values are used for initial organic and inorganic aerosol concentrations, to serve as a condensation core. To keep the particle dynamic as simple as possible, only one size section (representing aerosol with diameters between 0.1585 $\mu$m and 0.4 $\mu$m) is used. The simulation is run for about 5 hours with outputs every minute.

The different chemical schemes are combined with the CB05 mechanism for oxidant and $O_3$ formation. Four sets of configurations with SOA schemes of different complexity are studied here. In each setup, the scheme CB05 is supplemented with different SOA schemes for monoterpenes, sesquiterpene, toluene, and naphthalene. The SOA schemes are sumarized in Table 2. In the first setup, the SOA schemes represent near-explicit mechanisms. In the second setup, the SOA schemes of toluene and monoterpenes are schemes reduced using GENOA. The third and fourth setups involve SOA schemes from the

H$^2$O mechanism built from chamber experiments, with the third setup including the simple autoxidation scheme of Chrit et al. (2017) for monoterpenes. For monoterpenes, the near-explicit scheme is from MCM and PRAM (Peroxy Radical Autoxidation Mechanism) (Roldin et al., 2019), and the intermediate scheme is the GENOA reduced mechanism from Wang et al. (2023b). For toluene, the near-explicit scheme is from Lannuque et al. (2023). It is supplemented with the molecular rearrangement with ring opening of a bicyclic peroxy radical (BPR) with an O-O bridge (Iyer et al., 2023), as implemented in Sartelet et al. (2024),

and the GENOA reduced scheme is from Sartelet et al. (2024). For naphthalene, the near-explicit scheme is from Lannuque

| Precursor | Mechanism name | Type | Reference |
|---|---|---|---|
| Naphthalene | $H^2O$ | Implicit | Couvidat et al. (2013a) |
| | Expl. | Quasi-explicit | Lannuque and Sartelet (2024) |
| Toluene | $H^2O$ | Implicit | Couvidat et al. (2012) |
| | Expl. | Quasi-explicit with | Lannuque et al. (2023) |
| | | ipso-BPR molecular rearrangement | + Sartelet et al. (2024) |
| | Rdc | GENOA Reduced | Sartelet et al. (2024) |
| Monoterpenes | $H^2O$ | Implicit | Couvidat et al. (2012) |
| | Expl. | Quasi-explicit (MCM + PRAM) | Roldin et al. (2019); Wang et al. (2023b) |
| | Rdc | GENOA Reduced | Wang et al. (2023b) |
| $\beta$-caryophyllene | $H^2O$ | Implicit | Couvidat et al. (2012) |
| | Expl. | Quasi-explicit (MCM) | - |

**Table 2.** SOA chemical mechanisms used in the test cases of section 3.2

and Sartelet (2024). For sesquiterpene ($\beta$-caryophyllene), the near-explicit scheme is from MCM. Differences in the simulated concentrations and their temporal evolution among the implicit, quasi-explicit and GENOA-reduced schemes arise from the richer formation pathways included in the quasi-explicit and GENOA-reduced mechanims.

The toluene SOA concentrations simulated with the different configurations and with the different initial $NO_2$ conditions are shown in Fig. 4. The toluene SOA concentrations increase with time. Using the near-explicit schemes, the toluene SOA concentrations are lower when the initial $NO_2$ concentrations are decreased by a factor 2. If the initial $NO_2$ concentrations are increased by a factor 2, the toluene SOA concentrations are lower in the first 3 hours of the simulations, but they are higher after. As shown in the left panel of Fig. 4, the toluene SOA concentrations of the reduced and near-explicit schemes are very similar. However, as shown in the right panel of Fig. 4, the toluene SOA concentrations of the $H^2O$ scheme are lower than those of the near-explicit schemes. This was already observed in Sartelet et al. (2024). This is partly attributed to the impact of molecular rearrangement with ring opening of a BPR with an O-O bridge present in the near-explicit scheme, but which is missing in the $H^2O$ scheme. The evolution of the toluene SOA concentrations is similar between $H^2O$ and the near-explicit schemes when initial $NO_2$ concentrations are reduced: the toluene SOA concentrations decrease. SOA concentrations also decrease with the $H^2O$ scheme when initial $NO_2$ concentrations are doubled. This decrease persists through the whole simulation, contrary to the near-explicit scheme.

The monoterpene SOA concentrations simulated with the different schemes and in the different initial $NO_2$ conditions are compared in Fig. 5. With all mechanisms, monoterpene SOA concentrations increase as initial $NO_2$ levels decrease, because of the NOx regime, as already noted in Wang et al. (2024). The higher concentrations may be attributed to $RO_2$-$HO_2$ reaction rates, which tend to yield more highly oxidized monoterpene degradation products. As shown in the left panel of Fig. 5, the SOA concentrations simulated with the reduced schemes and their evolution with $NO_2$ levels are similar between the reduced and the near-explicit schemes. As shown in the lower panel of Fig.5, the monoterpene SOA concentrations are significantly

lower with the $H^2O$ mechanism that ignores autoxidation. Furthermore, monoterpene SOA concentrations decrease as $NO_2$ levels decrease in constrast to the simulation with the near-explicit mechanism. Taking into account autoxidation in the $H^2O$ mechanism leads to higher monoterpene SOA concentrations (upper right panel of Fig. 5), but they are still underestimated compared to the near-explicit scheme. Similarly to the near-explicit scheme, the $H^2O$ mechanism including autoxidation gives higher SOA concentrations with lower $NO_2$ levels.

The naphthalene SOA concentrations simulated with the different configurations and in the different $NO_2$ initial conditions are shown in the left panel of Fig. 6. The naphtalene SOA concentrations increase as the initial $NO_2$ levels increase. The naphtalene SOA concentrations simulated with the $H^2O$ scheme are much lower than those simulated with the near-explicit scheme. The naphtalene SOA concentrations simulated with the $H^2O$ scheme can even be considered negligible, except when the $NO_2$ levels are doubled. The difference between the concentrations simulated with the near-explicit and the $H^2O$ scheme reflect large uncertainties in the $H^2O$ scheme, which was built by (Couvidat et al., 2013a) from a specific set of experiments from (Chan et al., 2009) that are not representative of the full range of atmospheric conditions. Even though these experiments were corrected from wall-loss effects of particles, they were not corrected for gas wall losses. Furthermore, the behaviour of the $H^2O$ scheme may not fully extrapolate to all experimental setups, especially flow tubes where residence times are much shorter than in smog chambers.

The sesquiterpene SOA concentrations simulated with the different configurations and in the different $NO_2$ initial conditions are shown in the right panel of Fig. 6. During the first two hours of the simulations, the sesquiterpene SOA concentrations increase with increasing $NO_2$ levels. This is not the case in the simulation with the $H^2O$ mechanism, where the SOA concentrations are higher with the reference $NO_2$ levels.

These test cases highlight the importance of near-explicit chemical schemes, which incorporate various chemical pathways for SOA formation. In contrast, simplified schemes based on chamber experiments may simulate different evolution patterns under changing environmental conditions compared to near-explicit schemes. However, the reduced schemes generated by GENOA, as demonstrated in the examples presented in this study, successfully replicate the evolution predicted by near-explicit schemes.

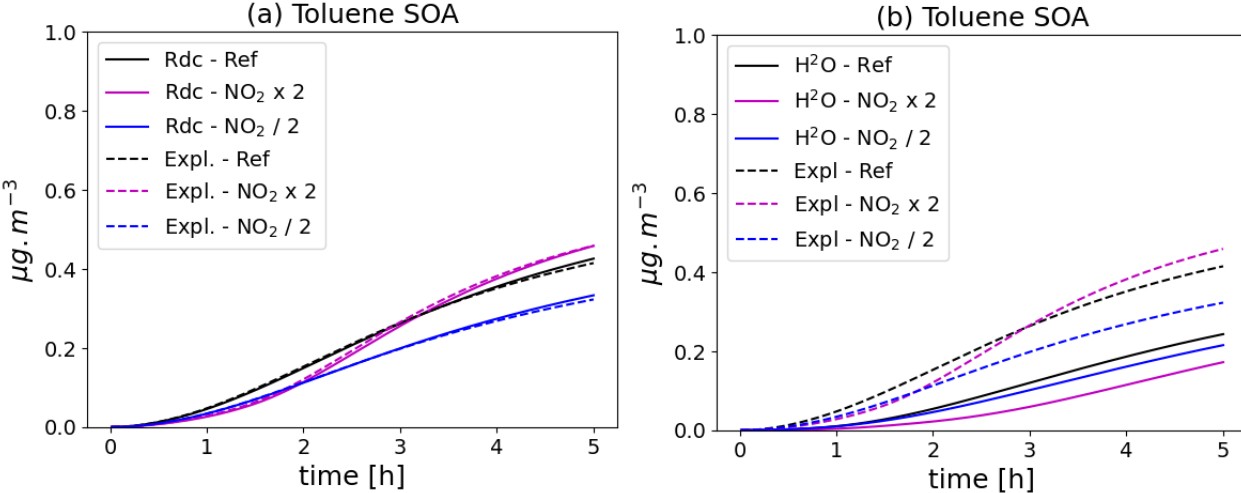

**Figure 4.** Time evolution of toluene SOA mass with different chemical schemes and for different $NO_2$ initial concentrations (Ref. denotes the reference $NO_2$ levels, $NO_2$ x 2 the simulation with doubled $NO_2$, and $NO_2$ / 2 the simulation with halved $NO_2$). In the right panel (a), the near-explicit chemical scheme of Lannuque et al. (2023) supplemented by molecular rearrangement with ring opening of a bicyclic peroxy radical with an O-O bridge Iyer et al. (2023); Sartelet et al. (2024) is compared to the $H^2O$ mechanism. In the left panel (b), the near-explicit scheme (Expl.) is compared to the reduced scheme (Rdc.) of Sartelet et al. (2024).

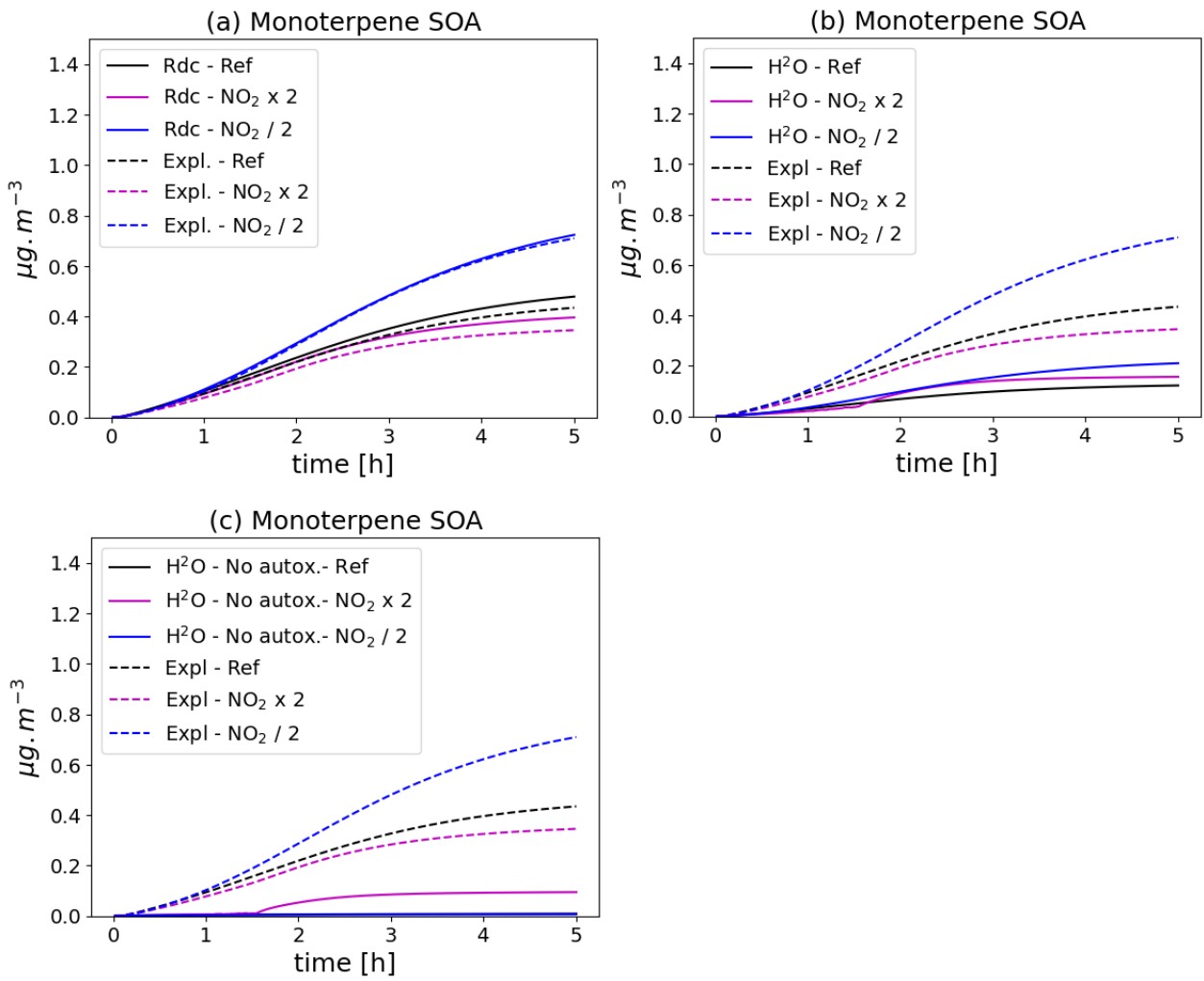

**Figure 5.** Time evolution of monoterpene SOA mass with different chemical schemes and for different NO$_2$ initial concentrations (Ref. denotes the reference NO$_2$ levels, NO$_2$ x 2 the simulation with doubled NO$_2$, and NO$_2$ / 2 the simulation with halved NO$_2$). In the upper left panel (a), the near-explicit chemical scheme (Expl.) of MCM and PRAM is compared to the reduced scheme (Rdc.) of Wang et al. (2023b). The near-explicit scheme is compared to the H$^2$O mechanism in the upper right panel (b) and to the H$^2$O mechanism without autoxidation in the lower panel (c).

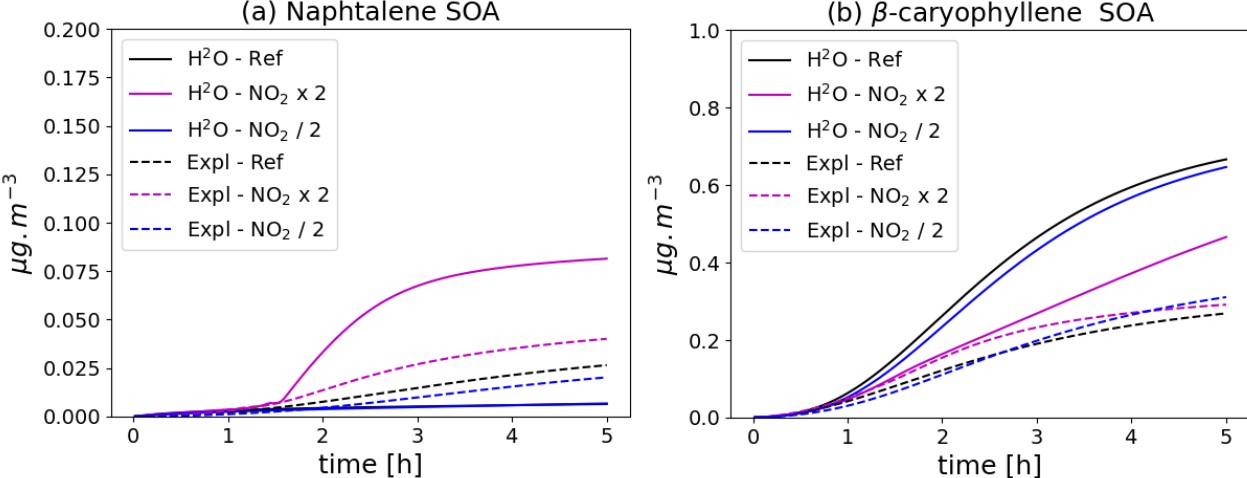

**Figure 6.** Time evolution of naphthalene (left panel, a) and $\beta$-caryophyllene (right panel, b) SOA mass with different chemical schemes and for different $NO_2$ initial concentrations (Ref. denotes the reference $NO_2$ levels, $NO_2$ x 2 the simulation with doubled $NO_2$, and $NO_2$ / 2 the simulation with halved $NO_2$). The near-explicit chemical scheme (Expl.) of Lannuque and Sartelet (2024) is compared to the $H^2O$ mechanism.

## 3.3 Comparisons to flow-tube or chamber experiments

### 3.3.1 Wall losses

To allow comparison of concentrations to flow-tube or chamber experiments, a wall loss module accounting for losses of particles and organic vapors was added in v2.0.

**Wall losses of organic vapors**

A reversible loss of organic vapors is accounted for SVOC species defined in the species list of aerosol with the following reaction from Huang et al. (2018):

$$\frac{dC_g}{dt} = -k_w^g (C_g - \frac{C_w}{K_w C_{wall}}) \tag{1}$$

with $C_g$ the gas phase concentrations, $k_w^g$ the wall losses kinetic of organic vapors, $C_w$ the equivalent wall concentration and $K_w$ the wall partitioning constant. $C_{wall}$ (in $\mu g/m^3$) has to be defined in the namelist. If $C_{wall}$ is not defined or equal to 0, wall deposition of organic vapors is not taken into account in the model. One possible method to estimate $C_{wall}$ and $K_w$ is to use the method of Huang et al. (2018):

$$C_{wall} = 10\,800 \, \frac{S}{V} \tag{2}$$

with S/V the surface on volume ratio of the chamber, and

$$K_w = \frac{K_p}{\gamma^{wall}} \times \frac{M_m}{200} \tag{3}$$

with $M_m$, the molar mass, $\gamma^{wall}$ the wall activity coefficient, $K_p$ the ideal partitioning constant:

$$K_p = \frac{760 \times 8.202 \times 10^{-5}}{10^6 M_m P_i^0(T)} \tag{4}$$

with $P_i^0(T)$ the saturation vapor pressure at temperature T (calculated from the properties defined in *aerosol_species_list_file*. The wall activity coefficient is calculated by this formula calculated for Teflon film (fluorinated ethylene propylene):

$$\gamma^{wall} = 10^{3.299} K_p^{0.6407} \tag{5}$$

$k_w^g$ can be provided directly by the user. Alternatively, $k_w^g$ can be calculated based on the eddy diffusion coefficient $k_e$ and the S/V ratio with the following formula:

$$k_w^g = \frac{S}{V} \left( \frac{\pi}{2\sqrt{k_e D_g}} + \frac{4}{\alpha_w \bar{c}} \right)^{-1} \tag{6}$$

with $D_g$ the gas-phase diffusion coefficient, $\bar{c}$ the mean speed velocity of the compound and $\alpha_w$ the accommodation coefficient for condensation onto the wall (Lannuque et al., 2023):

$$\alpha_w = 10^{-2.744} K_p^{-1.407}. \tag{7}$$

**Wall losses of particles**

The wall losses of particles are irreversible and are represented as:

$$\frac{dC_p}{dt} = -k_w^p C_p \tag{8}$$

with $C_p$ the particle phase concentrations, $k_w^p$ the wall losses kinetic of particles.

$k_w^p$ can either be provided directly by the user by defining the value of the parameter or calculated for each size bin. In that case, the user has to provide the eddy diffusion coefficient $k_e$, the S/V ratio, the radius of the chamber $R_{chamber}$ and the minimal wall loss rate $k_{wp0}$ due to electrostatic forces. $k_w^p$ is calculated based on Pierce et al. (2008):

$$k_w^p = k_{wp0} + \frac{6\sqrt{k_e D}}{\pi R_{chamber}} D_1 \left( \frac{\pi v_s}{2\sqrt{k_e D}} \right) + \frac{v_s}{4R_{chamber}/3} \tag{9}$$

with $D_1$ is the Debye function, D the Brownian diffusivity of the particle (dependent on the aerosol diameter) and the settling velocity of the particle (dependent on the aerosol diameter).

This wall-loss module is used in the following test cases that describe comparison of SOA formation in flow-tube or chamber experiments.

### 3.3.2 SOA formation from $\alpha$-pinene oxidation

This test case (illustrated in Fig. 7) is based on the work presented in Wang et al. (2023b). The two simulations represent the formation of SOA in an idealized oxidative flow reactor (OFR) Xavier et al. (2019) from the ozonolysis of $\alpha$-pinene. The oxidation of $\alpha$-pinene is simulated using the near-explicit mechanism from MCM and PRAM, and the mechanism reduced with GENOA. The saturation vapor pressure is calculated using UManSysProp (Topping et al., 2016), with the method of Myrdal and Yalkowsky (1997) to compute saturation vapor pressures and the method of Nannoolal et al. (2004) for the boiling point. The SOA concentrations and their temporal evolution are similar with the near-explicit and the reduced schemes. The quantity of SOA formed at the end of the simulation corresponds well to that measured by Xavier et al. (2019) (error below 4%).

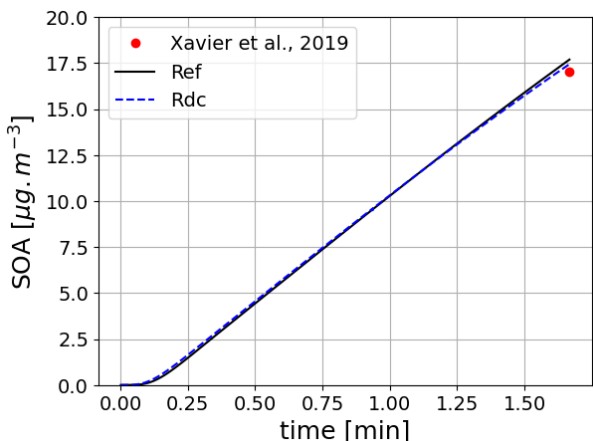

**Figure 7.** Time evolution of monoterpene ($\alpha$-pinene) SOA mass using the near-explicit mechanism (Ref) and the reduced mechanism (RdC). The results are compared to the measurements reported in Xavier et al. (2019).

### 3.3.3 SOA formation from toluene and naphthalene oxidation under experimental conditions

For toluene oxidation, the test case is illustrated in Fig. 8). It is based on the work of Lannuque et al. (2023). It represents the formation of SOA in an UV-lamp irradiated aerosol flow tube into which ammonium sulfate seeds and a high amount of gaseous toluene and isopropylnitrite (IPN) are injected. Toluene oxidation is here represented with the detailed mechanism of Lannuque et al. (2023), with the addition of the molecular rearrangement with ring opening of a bicyclic peroxy radical with an O-O bridge (ipso-BPR, Iyer et al. (2023); Sartelet et al. (2024)). Wall losses for stable gaseous species and the irreversible condensation of glyoxal and methylglyoxal are taken into account. IPN chemistry and experimental conditions representation are described in Lannuque et al. (2021). The dynamic of the condensation/evaporation on both organic and aqueous phases is solved without assuming thermodynamic equilibrium. The differences between the simulations presented in Fig. 8a lie in the simulation of the interactions in the particulate phase: ideal case (activity coefficients equal to 1), or interactions between uncharged molecules only (short range activity coefficients computed with UNIFAC), or interactions between uncharged molecules and

inorganic ions in aqueous phase (short, medium and long range activity coefficients computed with AIOMFAC). In the test-case conditions, considering interactions between uncharged molecules enhances the condensation while considering those with inorganic ions in the aqueous phase limits it, almost offsetting the first interactions. The SOA concentrations simulated after about 13 min compare reasonably well to the observed one, i.e. 16 $\mu$g m$^{-3}$.

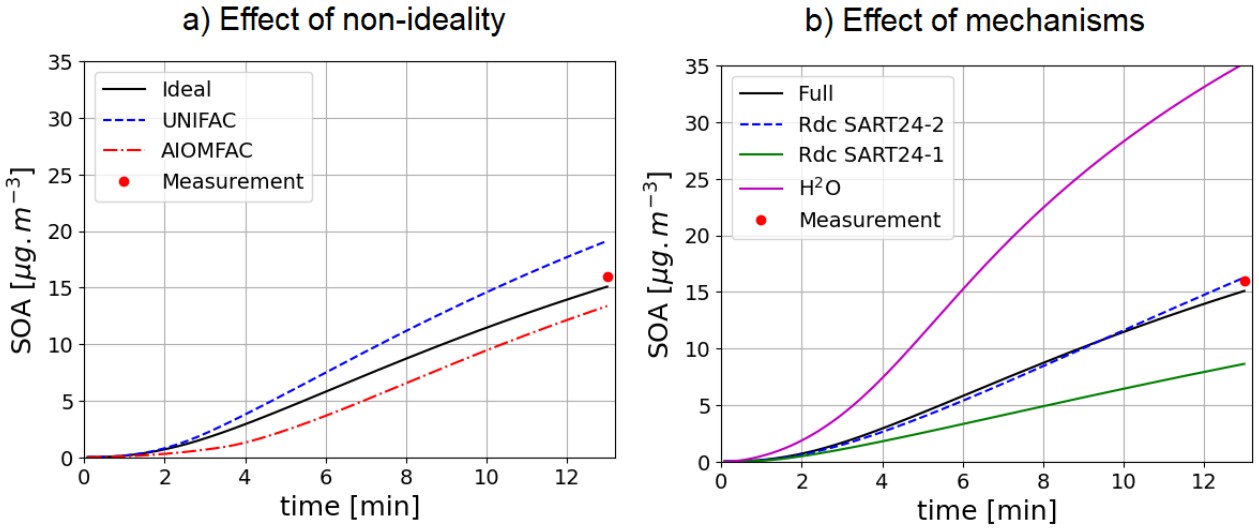

**Figure 8.** Time evolution of toluene SOA mass under experimental conditions taking into account or not interactions in condensed phases (a, left) or using different SOA mechanisms under ideal conditions (b, right).

The near-explicit toluene SOA mechanism of Lannuque et al. (2023) is replaced by reduced toluene SOA mechanisms generated with GENOA Sartelet et al. (2024). Three reduced toluene SOA mechanisms are presented in Fig. 8b. The first mechanism is reduced from MCM ("SART24-1"). The second one ("SART24-2") is the reduced mechanism of Lannuque et al. (2023), to which is added the molecular rearrangement with ring opening of a bicyclic peroxy radical with an O-O bridge (ipso-BPR, Iyer et al. (2023)). The third mechanism is the basic H$^2$O mechanism built from chamber experiments. These three mechanisms give similar results, although the mechanism reduced using the MCM ("SART24-1") tends to underestimate the concentrations. The H$^2$O scheme significantly overestimates the observed concentrations.

A similar test case was also created for naphthalene SOA based on the work of Lannuque and Sartelet (2024) with conditions similar to those of toluene. In this test case, illustrated in Fig. 9, considering the interactions and/or the viscosity in the particle limits the condensation of semi-volatile compounds, leading to less SOA formation than observed in the experiment (6.6 $\mu$g m$^{-3}$). When accounting for viscosity, diffusion inside the particles limits the condensation of semivolatile species because the particles are semisolid. However, both simulations with and without viscosity remain very close to the experimental data, one slightly overestimating and the other slightly underestimating, so this difference is not considered meaningful in regard of uncertainties on simulation parameters (such as the initial diameter of particles). For toluene SOA, viscosity has only

a minor influence, likely because their oxidation products are less multifunctional. The calculation of viscosity is detailed in the next section.

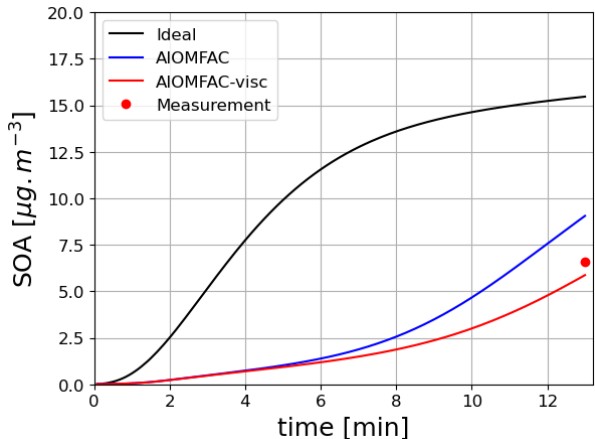

**Figure 9.** Time evolution of naphthalene SOA mass under different experimental conditions taking into account or not interactions and viscosity in condensed phases.

## 4  Advance gas/particle partitioning taking into account viscosity and particle-phase reactions

As illustrated in the test cases of the previous sections, organic compounds can condense on both an organic and an aqueous
phase. The condensation may be altered by the viscosity of the particles, as well by particle-phase reactions. The calculation of viscosity as a function of the particle composition, and the types of intra-particle reactions are now described.

### 4.1  Revision of the numerical solver

In order to represent diffusion inside viscous particles, those are discretized into layers. To explicitly represent this diffusion, a high number of layers would be necessary for the model to be accurate. To reduce the number of layers, alternative ap-
proaches involve the use of an effective mass accommodation coefficient (Shiraiwa and Pöschl, 2021; Lakey et al., 2023), or a methodology that represents implicitly the diffusion inside particles (Couvidat and Sartelet, 2015). The latter method is used in SSH-aerosol v2.0. As in Couvidat and Sartelet (2015) and in SSH-aerosol v1.1, it does not represent the exchange of compounds between layers but assumes a characteristic time to reach the interface. As illustrated by Fig. 10, diffusion is represented by exchange fluxes between the different layers inside the particle and the interface layer. The main consequence of this
simplification is that the diffusion of a compound is computed assuming identical affinity across all layers, where this affinity represents the compound's activity relative to the other particle-phase constituents. It may therefore miss some entrapment effect (impossibility for compounds to cross a layer because of a lack of affinity).

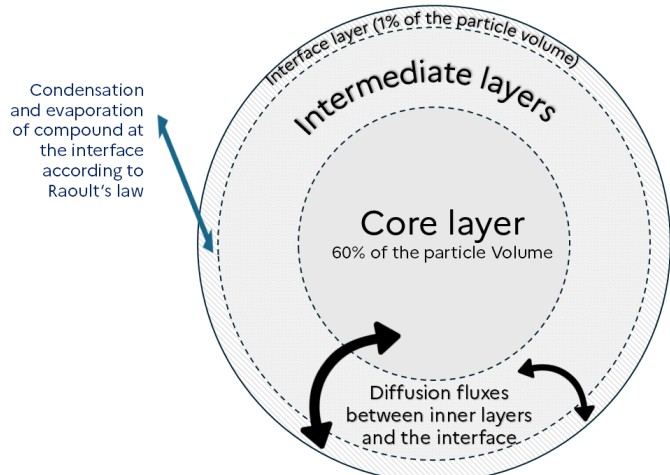

**Figure 10.** Illustration of implicit treatment of diffusion inside particle.

The flux of diffusion for a specific layer, bin and species ($J_{\text{diff},i}^{\text{bin,layer}}$ in $\mu$g m$^{-3}$ of air s$^{-1}$) is calculated as:

$$J_{\text{diff},i}^{\text{bin,layer}} = k_{\text{diffusion}}^{\text{bin,layer}} \left( A_{p,i}^{\text{interface,layer}} \frac{K_{p,i}^{\text{bin,layer}} M_o^{\text{bin,layer}}}{K_{p,i}^{\text{bin,interface}} M_o^{\text{bin,interface}}} - A_{p,i}^{\text{bin,layer}} \right) \tag{10}$$

with $k_{\text{diffusion}}^{\text{bin,layer}}$ the parameter of diffusion rate from the interface to the layer (in s$^{-1}$), "interface" the index of the layer at the gas/particle interface, $K_{p,i}^{bin,layer}$ (in m$^3$ $\mu$g$^{-1}$) the partitioning constant, $M_o^{bin,layer}$ (in $\mu$g m$^{-3}$ of air) the mass of the organic phase in the layer and $A_{p,i}^{bin,layer}$ the particle concentrations of the species i in the layer (in $\mu$g m$^{-3}$ of air).

An hybrid explicit numerical method is used to solve condensation/evaporation/diffusion in Couvidat and Sartelet (2015) and SSH-aerosol v1.1. One issue with this method is the need to distinguish cases where a compound condenses or evaporates

very rapidly (typically handled with an equilibrium approach) from cases that require a dynamic resolution. The numerical method is replaced by an iterative implicit numerical (backward Euler), which is capable of simultaneously representing both types of cases:

$$B_{p,i}^{\text{bin,layer}}(t+\Delta t, iter+1) = \frac{A_{p,i}^{\text{bin,layer}}(t) + Prod_i^{\text{bin,layer}}(t+\Delta t, iter)\Delta t}{1 + k_{\text{loss},i}^{\text{bin,layer}}\Delta t} \tag{11}$$

With iter the iteration index, $A_{p,i}^{\text{bin,layer}}$ the initial concentration of compound i in a specific bin and layer, $B_{p,i}^{\text{bin,layer}}$(t+$\Delta$

480   t,iter+1) the restimated concentrations, $Prod_i^{\text{bin,layer}}$ the production rate and $k_{\text{loss},i}^{\text{bin,layer}}$ the loss kinetic rate. $Prod_i^{\text{bin,layer}}$ and $k_{\text{loss},i}^{\text{bin,layer}}$ include the production and loss due to diffusion, chemical reactions as well condensation/evaporation for the layer at the interface with air.

This method is iterative, as $Prod_i^{\text{bin,layer}}$ is determined for concentrations at time t+$\Delta$t. Iteration are performed until convergence is reached with the iterative solver described in Couvidat and Sartelet (2015):

$$A_{p,i}^{\text{bin,layer}}(t+\Delta t, iter+1) = f \times B_{p,i}^{\text{bin,layer}}(t+\Delta t, iter+1) + (1-f) \times A_{p,i}^{\text{bin,layer}}(t+\Delta t, iter) \tag{12}$$

where $B_{p,i}^{bin,layer}(t+\Delta t)$ is the concentration estimated with Eq. 11 and f is a weighting factor ranging from 0 to 1 determined to ensure convergence. Following Couvidat and Sartelet (2015), f is divided by a factor 2 (therefore decreasing the variations of concentrations between two iterations) each time a non-convergence loop is detected to ensure convergence.

The implicit solver is found to be faster and more stable than the previous method based on the explicit solver. Note that the same solver is applied for aqueous-phase concentrations (but with only one layer and without representing diffusion inside the aqueous phase).

## 4.2 Viscosity

The gas/particle mass transfer is strongly affected by particle viscosity, modeled by diffusion coefficients that are used to compute $k_{diffusion}^{bin,layer}$ such as:

$$k_{diffusion}^{bin,layer} = \frac{1}{\tau_{dif}\alpha_{layer}} \tag{13}$$

$$\tau_{dif} = \frac{R_p}{\pi^2 D_{org}} \tag{14}$$

with $R_p$ the radius of the particle, $D_{org}$ the diffusion coefficient inside the particle and $\tau_{dif}$ the characteristic time of diffusion. The higher the viscosity is, the lower the diffusion coefficient is. To illustrate this effect, in SSH-aerosol v1.1, the condensation of hydrophobic organic surrogates of different volatility was studied for different values of viscosity.

In SSH-aerosol v2.0, the viscosity may be estimated as a function of the particle composition, with the AIOMFAC-VISC algorithm Gervasi et al. (2020). In this algorithm, the UNIFAC parameters are used to compute the viscosity. Therefore, the viscosity of a mixture of organic compounds can be computed as long as the decomposition in functional groups of the compounds is given. It should be noted that the aqueous-phase is considered to be inviscid. In the organic phase, by allowing the diffusion coefficient to vary with viscosity, $D_{org}$ is no longer constant and therefore Eq. 13 is rewritten, as a function of the apparent diffusion coefficient from the interface to the considered layer $D_{org,app,i}^{bin,layer}$:

$$k_{diffusion}^{bin,layer} = \frac{\pi^2 D_{org,app,i}^{bin,layer}}{R_p\alpha_{layer}} \tag{15}$$

$D_{org,app,i}^{bin,layer}$ is computed with AIOMFAC-VISC by using cumulated concentrations between the interface and the considered layer in order to provide an average diffusion coefficient.

Following DeRieux et al. (2018), in the AIOMFAC-VISC algorithm, a fragility parameter $D_{fragility}$ is used to compute pure compound viscosity $\eta_0$:

$$log_{10}(\eta_0) = -5.0 + 0.434\frac{D_{fragility}*T_0}{T-T_0} \tag{16}$$

with $T_0$ the Vogel temperature, or ideal glass transition temperature:

$$T_0 = \frac{T_g*39.17}{D_{fragility}+39.17} \tag{17}$$

with $T_g$ the glass transition temperature, a characteristic temperature (computed as a function of the number of N, C, and O atoms) at which the mixture changes from a glassy solid state to a semi-solid state. Although $D_{fragility}$ is set to 10 in Gervasi et al. (2020), a parameterisation of $D_{fragility}$ as a function of the O/C ratio is used, following Schmedding et al. (2020). A

The diffusion coefficient $D_{diff_c}$ of the compound $c$ is deduced from the viscosity ($\eta$), the viscosity of pure water ($\eta_0$) and the diffusion coefficient in pure water $D_{diff,water}$, following the parameterization of Maclean et al. (2021):

$$D_{diff_c} = D_{diff,water} \times \left(\frac{\eta_0^{water}}{\eta}\right)^{\xi} \tag{18}$$

where $\xi$ is a parameter used to compare the radius of the diffusing molecule ($R_{diff}$) to the average radius of the compound mixture:

$$\xi = 1 - A\, exp\left(B\frac{R_{diff}}{R_{mean}}\right) \tag{19}$$

with A and B equal to 0.73 and 1.79 respectively (Maclean et al., 2021), and $R_{mean}$ is calculated as the average value of $R_{diff}$ weighted by molar fractions. $R_{diff}$ is calculated from the density and molar mass of the compound:

$$R_{diff} = \left(\frac{3M_c}{4N_a\rho_c * 1000}\right)^{\frac{1}{3}} \tag{20}$$

with $M_c$ the molar mass (in g/mol), $N_a$ the Avogadro number and $\rho_c$ the compound density (in kg/m$^3$).

This methodology allows to calculate the diffusion coefficients for each species and each particle layer and therefore to vary with particle composition. The effect of changing composition on the condensation of a semivolatile compound is illustrated in Fig. 11. The condensation of an organic compound (POAmP) is investigated for different particle compositions, and thus for different viscosities. POAmP is represented using the default molecular structure proposed by Couvidat et al. (2012) in the H$^2$O mechanism for primary compounds and model species with unknown molecular structures. In this representation, the activity coefficient is set to unity, ensuring that the condensation process is not influenced by non-ideality effects. At the start of the simulation, a low concentration of POAmP in the gas phase (0.01 $\mu$g/m$^3$) was set in order to not affect the viscosity of the organic phase. The time evolution of POAmP concentrations was computed for different relative humidity (RH) and for different compositions of the aerosol onto which POAmP condenses. The compositions differ by the number of alcohol groups (OH), of carbons, and of various types of functional groups (ketone, hydroxyl, acid, and nitrate). The simulations show that an aerosol composed of compounds with 16 carbons and 4 alcohol groups is very viscous at RH=10% and that condensation of POAmP hardly occurs. However, at RH=70%, the aerosol can be considered as inviscid and the condensation of POAmP is non-limited by viscosity. Adding just one OH group has a strong effect on viscosity at RH=10%. With 16 carbons and 2 OH groups, equilibrium between the gas and particle phases of POAmP is reached in about one hour. However, it takes more than 100 hours to reach equilibrium with an aerosol composed of 16 carbons and 3 OH groups. Adding, acid, hydroperoxide or nitrate groups affects the viscosity even more significantly. Especially, just by adding one nitrate group, the aerosol switches from inviscid to extremely viscous. By comparison, the effect on viscosity of increasing the number of carbons is low. If the molecules have 2 OH groups, equilibrium is reached in less than 1 hour with less than 16 carbons. Time to equilibrium increases progressively with the number of carbons. It takes about 100 hours with 20 carbons.

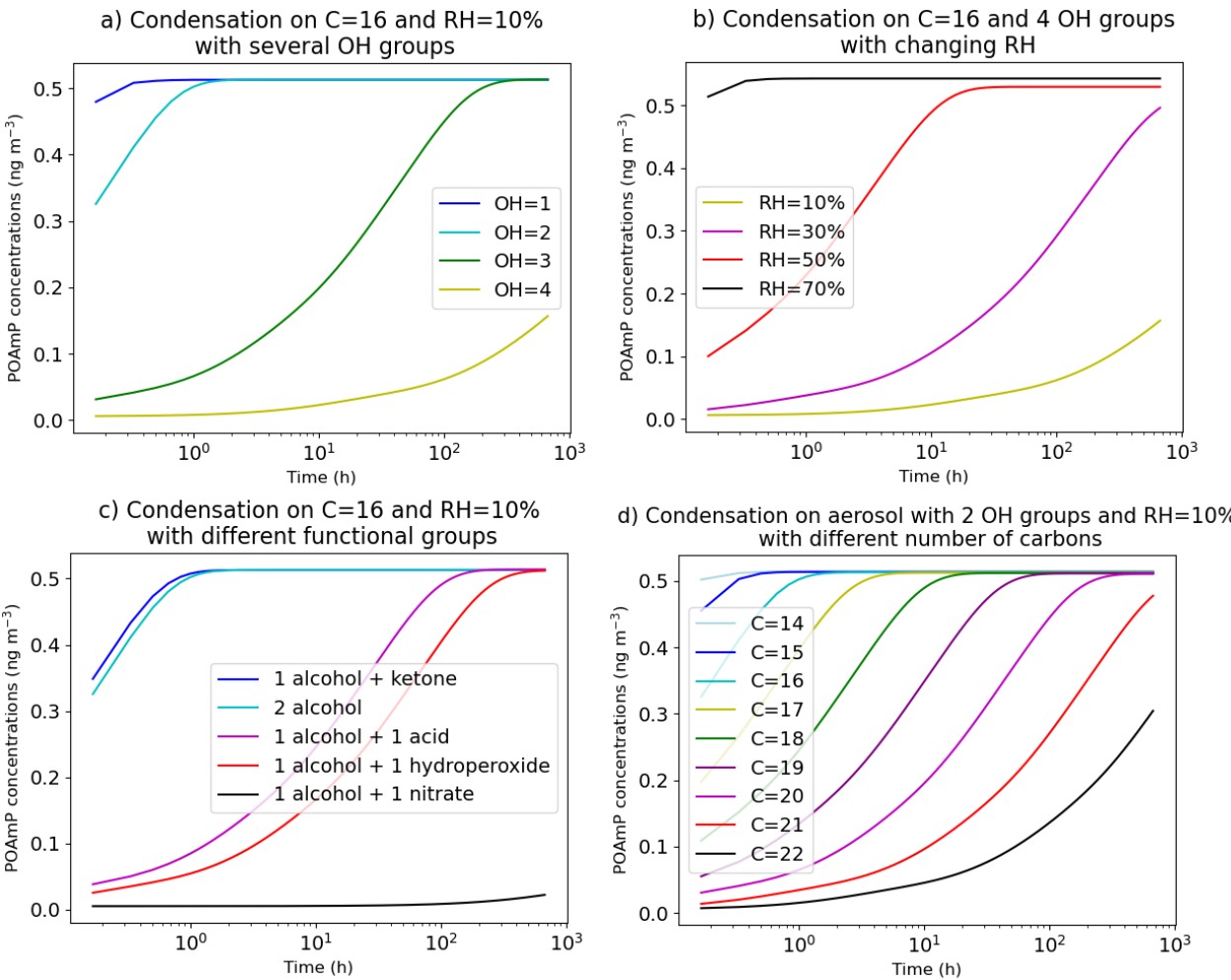

**Figure 11.** Concentration of POAmP condensing onto 5 $\mu$g/m$^3$ of a compound with 16 carbons C$_{16}$ as a function of time. The different graphs show the effect of changing the composition of the aerosol or humidity.

### 4.3 Particle-phase reactions

Reactions can occur inside the particles. These reactions can be activated or deactivated by modifying the file indicated by *reaction_soap_file* in the section &physic_condensation of the namelist. Different types of reactions can be accounted for. Examples on how to use this reaction file are provided in the SSH-aerosol guide (Sartelet et al., 2025a). Currently 5 types of particle-phase reactions are taken into account: $1^{st}$ order irreversible reaction, $2^{nd}$ order reversible oligomerization, hydratation of aldehydes, reactions of organic compounds with inorganic ions and a specific paramterization for pinonaldehyde oligomerization.

### 4.3.1  $1^{st}$ irreversible order reaction

An irreversible $1^{st}$ order reaction is a reaction that transforms X into Y is represented with a kinetic parameter k such as:

$$X \xrightarrow{k} Y \tag{21}$$

This type of reaction can account for catalysis by water and pH. It allows to set a lifetime of semi-volatile compounds in particles, e.g. a lifetime of 1 h corresponds to a kinetic rate of about 3 $10^{-4}$ s$^{-1}$. This type of reaction could also be used to
represent hydrolysis (for example the hydrolysis of nitrate groups into hydroxyl groups).

### 4.3.2  Type 1: Bulk oligomerization

In most modeling studies, a parameterization for oligomerization based on a simple first-order irreversible reaction is used (Carlton et al., 2008; Lemaire et al., 2016). However, most oligomerization pathways should involved second-order reversible reactions (like esterification, hemiacetalization, aldolization, peroxyhemiacetalization) catalyzed under acidic conditions and
unfavored by humid conditions Couvidat et al. (2018b).

In SSH-aerosol v2.0, oligomerization is based on the parameterization developped by Couvidat et al. (2018b). It is represented by a reversible process, unfavored by humid conditions, with a single reaction estimating if a compound is present as a monomer or as an oligomer. Oligomers from one species are therefore represented by a single oligomer species. The parameterization is termed "Bulk oligomerization" (as it does not explicitly represent all possible oligomers) and can be used to represent
oligomerization in a complex mixture of compounds. It can be schematized with the following reaction (A representing the monomer and $A_{oligo}$ representing any monomer block present inside the oligomers):

$$A \underset{k_{reverse}}{\overset{k_{oligo}}{\leftrightarrow}} \frac{1}{m_{oligo}} A_{oligo} \tag{22}$$

with A a monomer compound, $A_{oligo}$ the monomer blocks of compound A inside oligomers, $m_{oligo}$ the number of monomer blocks inside oligomers, $k_{oligo}$ the kinetic rate parameter of oligomerization and $k_{reverse}$ the kinetic rate parameter of the
reverse reaction.

The net flux of oligomerization $J_{oligo}$ is computed with the following equations:

$$J_{oligo} = -\frac{dX_{A,monomer}}{dt} = k_{oligo}a_{A,monomer} - k_{reverse}a_{A,oligomer} \tag{23}$$

with $X_{A,monomer}$ the molar fraction of compound A, $a_{A,monomer}$ the activity on a molar fraction basis of compound A and $a_{A,oligomer}$ the activity on a molar fraction basis of the oligomer formed from compound A.
The kinetic rate of oligomerization $k_{oligo}$ is computed as follows:

$$k_{oligo} = k_{oligo}^{max} \sum_i a_{i,monomer} \tag{24}$$

with $k_{oligo}^{max}$ the maximum kinetic rate parameter for oligomerization, and $\sum_i a_{i,monomer}$ the sum of monomer activities.

The reverse kinetic rate parameter is estimated from the equilibrium constant for oligomerization (due to esterification or a

similar oligomerization mechanism), which may be written as

$$(K_{oligo}^{eq})^{m_{oligo}-1} = \frac{a_{A,oligomer}(a_{H_2O})^{m_{oligo}-1}}{a_{A,monomer}(\sum_i a_{i,monomer})^{m_{oligo}-1}} \tag{25}$$

with $a_{H_2O}$ the activity of water on a molar basis, $m_{oligo}$ is the number of oligomer blocks in oligomers, $K_{oligo}^{eq}$ is the the equilibrium oligomerization constant. $k_{oligo}^{max}$, $m_{oligo}$ and $K_{oligo}^{eq}$ have to be provided by the user for each reactions.

To illustrate the formation of oligomers, the Platt test case, describing the SOA formation from exhaust emissions from a Euro 5 gasoline car in a chamber (Platt et al., 2013), modified by considering isoprene and $NH_3$ emissions (Sartelet et al., 2020), is run, allowing isoprene SOA to form oligomers based on the parameters proposed by Couvidat et al. (2018a) ($k_{oligo}^{max}$ = 2.2 × 10⁻⁴ s⁻¹, $m_{oligo}$ = 3.35 and $K_{oligo}^{eq}$ = 2.94). The evolution of the concentrations of isoprene SOA, monomers, and oligomers in the particles as a function of time for these simulations is shown in Fig. 12. Although, the SOA concentration does not change significantly when oligomerization is activated, the SOA composition is strongly impacted. After 1 hour, almost all the SOA mass is constituted by monomer species. However, after 5 hours, monomer species constitute only 50% of isoprene SOA.

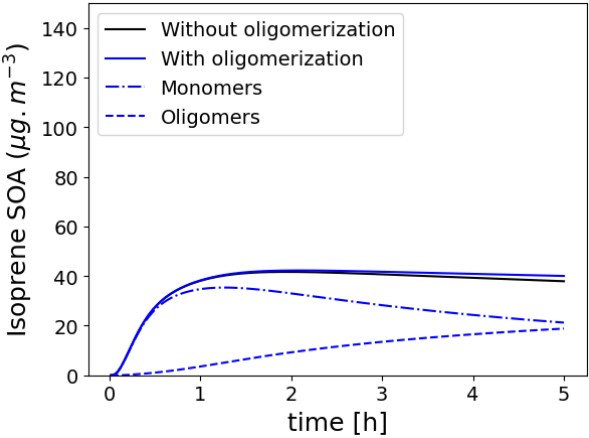

**Figure 12.** Time evolution of isoprene SOA, monomers, and oligomers in the particle phase.

### 4.3.3 Type 2: Hydratation of aldehydes

Aldehydes (and more generally carbonyl compounds) can undergo hydratation. In that case, the carbonyl group is transformed into a diol, and hydratation is represented as an equilibrium between two species, the aldehyde and the diol, using an hydratation constant $K_{hydratation}$:

$$\frac{[diol]}{[aldehyde]} = K_{hydratation}\frac{\gamma_{diol}^{\infty}\gamma_{aldehyde}}{\gamma_{diol}\gamma_{aldehyde}^{\infty}}a_w \tag{26}$$

with a$_w$ the activity of water (equal to RH at equilibrium). $\gamma_{diol}$ and $\gamma_{aldehyde}$ corresponds to the activity coefficient of the diol and the aldehyde. They are normalized by the activity coefficient at infinite dilution $\gamma^\infty$, as the hydratation constant is generally measured in the aqueous phase.

The equilibrium constant between the diol and the aldehyde is generally quite low (0.57 for butanal according to GECKO-A Camredon et al. (2007)). However, glyoxal can be hydradated twice with a very high constant as illustrated in Fig. 13.

**Figure 13.** Illustration of glyoxal conversion into polyols due to hydration. The hydratation constants are taken from Ervens and Volkamer (2010).

The influence of hydratation on glyoxal absorption was tested by running the Platt isoprene test case including ammonia emission with and without hydratation. While the concentrations of glyoxal in the particle phase without hydratation are negligible (below $10^{-8}$ $\mu$g/m$^3$), the total concentrations of glyoxal and its hydrate are higher and reach $1.8 \times 10^{-3}$ $\mu$g/m$^3$.

### 4.3.4 Type 3: Reaction of organic compounds with inorganic ions

Organic compounds in the aqueous can react with inorganic ions to form other compounds. For example, the compound IEPOX (epoxide formed by the oxidation of isoprene) can react with $SO_4^{2-}$ and $HSO_4^-$ ions to form organosulfate (Couvidat et al., 2013b) or with $NO_3^-$ to form organonitrate. "This type of reaction can account for catalysis by water (reaction rate multiplied by the activity of water) and pH (reaction rate multiplied by the activity of H$^+$ ion). The Fig. 14 illustrates the formation of SOA from IEPOX based on Couvidat et al. (2013b) due to intra-particle reactions for different amounts of ammonia, and hence different pH of the particle. In the absence of ammonia (very acidic particles), the formation of organosulfate is very important. For partially neutralized sulfate, the formation of organosulfate decreases but is compensated by the formation of methyltetrols due to the hydrolysis of IEPOX. When sulfate is almost totally neutralized, the concentrations of SOA formed by the hydrolysis of IEPOX decreases significantly and is strongly dominated by the formation of methyltetrols.

### 4.3.5 Type 4: Oligomerization of pinonaldehyde

Concerning organic reactions in the particles, oligomerization of pinonaldehyde (BiA0D, species in H$^2$O) may be considered by the user. It corresponds to the parameterisation detailed in Pun and Seigneur (2007); Couvidat et al. (2012), where BiA0D undergoes oligomerization via an effective Henry's law constant

$$H_{eff} = H \left(1 + 0.1 \left(\frac{a(H^+)}{10^{-6}}\right)^{1.91}\right) \tag{27}$$

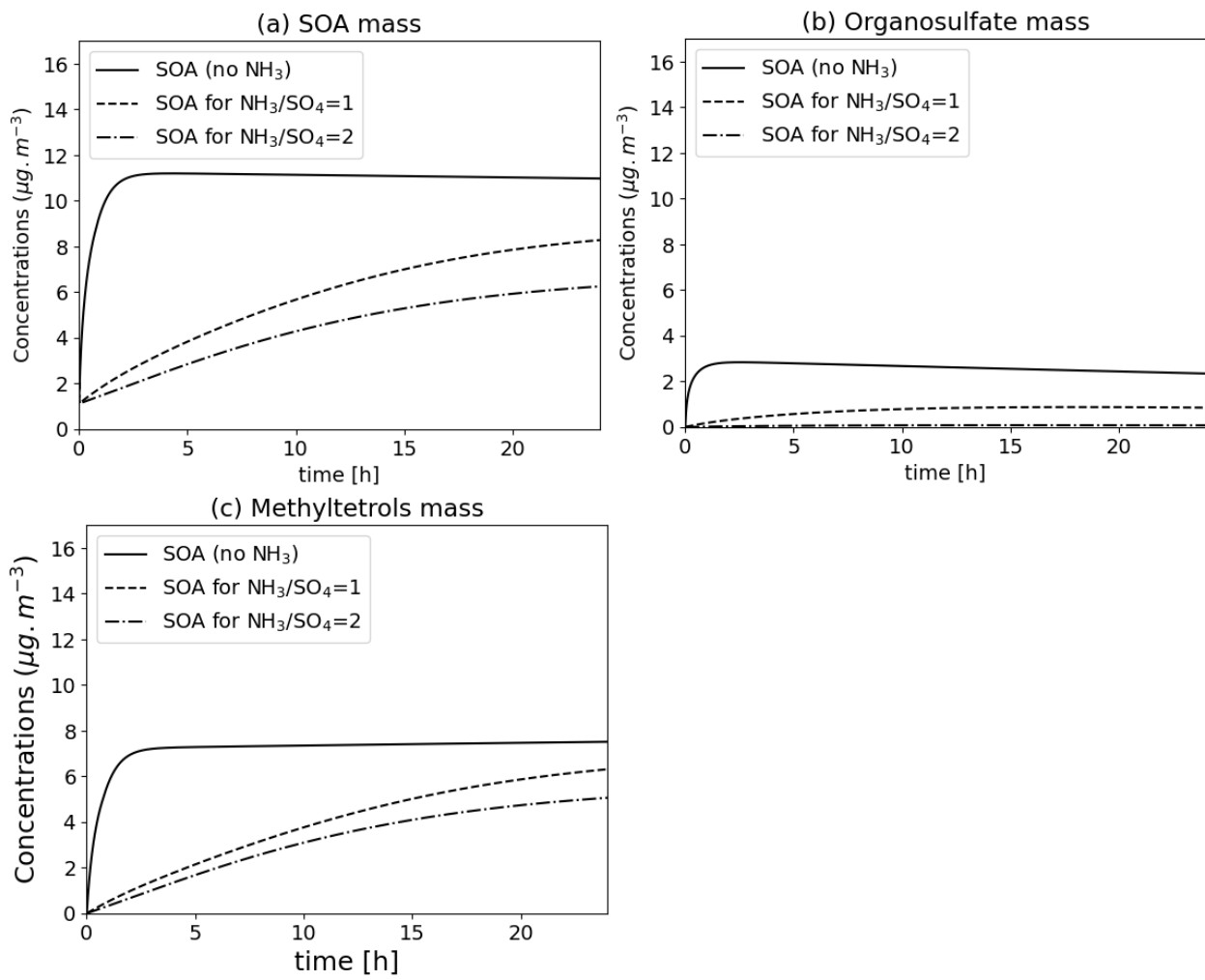

**Figure 14.** Formation of SOA (top, panel a), organosulfate (bottom left, panel b) and methyltetrols (bottom right, panel c) due to the reaction of 10 $\mu$g/m$^3$ IEPOX in the particle phase (in the presence of 10 $\mu$g/m$^3$ of HNO$_3$ at the beginning of the reaction and 3 $\mu$g/m$^3$ of sulfate) for a different ratio of ammonia to sulfate.

where $H_{eff}$ is the effective Henry's law constant of BiA0D, H is the monomer Henry's law constant of BiA0D, and a(H$^+$) is the activity of protons in the aqueous phase.

It should be noted that this parameterization (treating oligomerization with an equilibrium constant) has been initially developed to increase SOA formation from pinonaldehyde (Pun and Seigneur, 2007), and was present in SSH-aerosol v1.1. However, Couvidat et al. (2013b) showed that the uptake of pinonaldehyde on acidic particle is very slow and should not be treated with an equilibrium constant.

## 4.4   Irreversible condensation of methylglyoxal and glyoxal

Once uptaken into the aqueous phase of the particles, dicarbonyls can be oxidized by OH radicals present in this aqueous phase to irreversibly form highly soluble and low-volatility compounds (Hu et al., 2022). Empirical reactions representing this irreversible condensation pathway for glyoxal and methylglyoxal has been added in SSH-aerosol v2.0. The term "irreversible condensation pathway" is used in contrast to the classical "reversible condensation pathway" influenced by hydration and isomerization. The irreversible condensation pathway is represented by the following reaction:

$$DICARB \xrightarrow{k_{irdi}} IRDICARB \tag{28}$$

where DICARB is the semi-volatile species studied (glyoxal or methylglyoxal) and IRDICARB its non-volatile counterpart via irreversible partitioning. The kinetics $k_{irdi}$ of this reaction follows the parameterization of Curry et al. (2018):

$$k_{irdi} = \frac{1}{4} v \times \gamma \times A_{surf} \tag{29}$$

with:

$$v = \sqrt{\frac{8RT}{\pi M_{dicarb}}} \tag{30}$$

$$\frac{1}{\gamma} = \frac{1}{\alpha} + \frac{v}{4RTH_{dicarb}^{eff}\sqrt{k^l D_{aq}}} \times \frac{1}{\coth q - 1/q} \tag{31}$$

$$q = \frac{R_p}{\sqrt{\frac{D_{aq}}{k^l}}} \tag{32}$$

where $\gamma$ is the reactive uptake coefficient, $v$ and $M_{dicarb}$ are the gas-phase thermal velocity and the molar mass of the DICARB species, $A_{surf}$ is the aerosol surface, $R$ is the universal gas constant, $T$ is the temperature in Kelvins, $H_{dicarb}^{eff}$ is the effective Henry's law constant, $q$ is a parameter that account for in-particle diffusion limitation, $R_p$ is the particle radius, $k^l$ is the first order loss rate due to reaction with OH radicals in the aqueous phase. The aqueous-phase diffusion coefficient D$_{aq}$ is fixed to $10^{-9}$ m$^2$ s$^{-1}$, which is typical for small organics (Bird et al., 2006), and the mass accommodation coefficient $\alpha$ to 0.02, similar to that of formaldehyde uptake to water (Jayne et al., 1992). Henry constants of DICARB are adapted from Sander (2015)

to match with Hu et al. (2022) observations and the Henry constant of OH used to estimate OH aqueous concentration from gaseous ones is adapted from Sander (2015) leading to aqueous concentration in agreement with typical mean value in the atmosphere from Herrmann et al. (2010). Further details and description of the parameters are available in Curry et al. (2018).

## 4.5 Coupling between oganics and inorganics

The standard configuration of SSH-aerosol is based on two separate thermodynamic modules: SOAP and ISORROPIA. ISORROPIA is called first to compute water, pH, and inorganics. Then, SOAP is called to compute the concentrations of organics. In this system, organics do not affect the computation of pH or the concentrations of inorganics. This requires solve organics and inorganics together.

SOAP was modified in order to be able to solve the equilibrium and the dynamics of inorganic and organic aerosol simultaneously. Organics can affect the formation of inorganics by:

1. Influencing the condensation of water, which in turn affects the absorption of $HNO_3$ and $NH_3$.

2. Influencing the computation of pH as the dissociation of acid organics is accounted for.

3. Influencing the computation of molalities (concentrations of ions in mol/kg). In ISORROPIA, molalities are computed as mol/kg of water. In SOAP, all solvent molecules are accounted for (water but also organics).

SOAP is based on the AIOMFAC thermodynamic module that accounts for numerous ions, $H^+$, $OH^-$, $NO_3^-$, $HSO_4^-$, $SO_4^{2-}$, $HCO_3^-$, $CO_3^{2-}$, $Cl^-$, $Na^+$, $K^+$, $Mg^{2+}$, $Ca^{2+}$. Inorganic aerosols are assumed to be thermodynamically metastable (inorganic compounds are always present in the aqueous phase of particles and no solid salts are formed) with the exception of $CaCO_3$ that is assumed to have a very low solubility. Henry's law constant are taken from Fountoukis and Nenes (2007). The iterative solver in Eq. 12 is used to solve iteratively the system of equations. f in Eq. 12 is constrained in order to ensure that activity coefficients of inorganics and pH do not change significantly between two iterations in order to ensure stability and convergence.

Several test cases, such as the Platt test case with isoprene and $NH_3$, have been run using ISORROPIA or SOAP for inorganics. Using SOAP, the simulated concentrations are very similar to the results obtained using ISORROPIA, as shown in the guide (Sartelet et al., 2025a) To illustrate some of the capabilities of the SOAP model for both inorganic and organic thermodynamic, a test case simulates the evolution of $CaCO_3$ in the presence or in the absence of $HNO_3$ with an equilibrium approach or a dynamic approach. As shown in Fig. 15, when $CaCO_3$ is not exposed to $HNO_3$ the particle does not evolve. Especially, it does not absorb water. This is due to the fact that $CaCO_3$ is currently the only particle for which delisquescence is accounted for due to its very low saturation value that would correspond to a delisquescence relative humidity close to 100%. When [$Ca^{2+}$ (mol/kg of solvents)][$CO_3^{2-}$ (mol/kg of solvents)]>K (with K the equilibrium constant), a solid $CaCO_3$ is formed. When exposed to $HNO_3$, around 2 molecules of $NO_3^-$ will replace 1 molecule of $CO_3^{2-}$ to maintain electroneutrality. It leads to the presence of the nitrate and water inside the particle, and part of $Ca^{2+}$ that was previously associated with $CO_3^{2+}$ is transferred to the aqueous phase. Therefore, due to the condensation of $HNO_3$, the particle grows bigger in size and becomes hygroscopic.

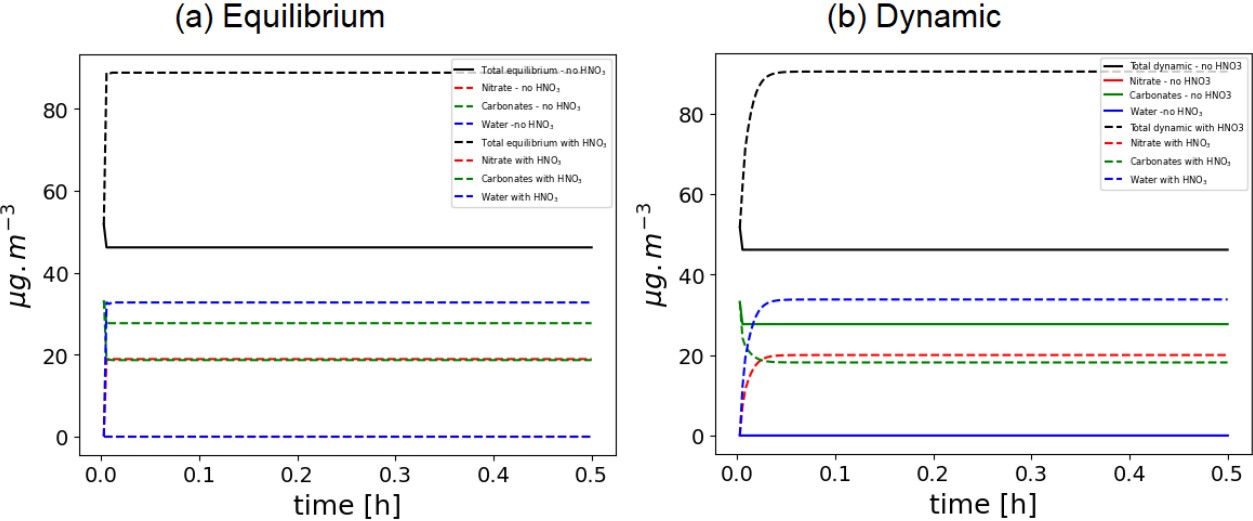

**Figure 15.** Time evolution of $CaCO_3$ particles in presence or in absence of $HNO_3$ with an equilibrium approach (a) or a dynamic approach (b).

## 5  Representation of ultrafine particle concentrations

Ultrafine particle (UFP) concentrations are strongly influenced by nucleation, coagulation and condensation of extremely-low
volatile compounds. For coagulation, the discretized equations governing aerosol dynamics involve partition coefficients that
account for the possibility that the coagulation of particles from two given size sections may produce particle sizes spanning
multiple sections. The size mesh is assumed to remain fixed over time, so that these partition coefficients can be pre-computed.
In previous model version, the coagulation partition coefficients were calculated using a Monte-Carlo approach. In v2.0, to
gain CPU time when calculating coagulation partition coefficients, an explicit formulation was derived for particles that are
internally mixed, as detailed in Jacquot and Sartelet (2024). If the mixing state is resolved, the coagulation partition coefficients
are solved, as in v1.1, i.e. using a Monte-Carlo approach. Because the explicit formulation has negligible computational cost
compared with the Monte Carlo approach, only the explicit formulation is retained for internally-mixed particles in v2.0.

Given the strong impact of coagulation on the growth of ultrafine particles, it should always be included in their modelling.
Consequently, the particle-diameter limits are held constant over the course of the simulations. Because nucleation corresponds
to the formation of nanometric particles from gas precursors, the lowest particle diameter bound of the size distribution should
be about 1 nm. Different parameterizations are implemented to represent nucleation, as detailed in section 5.1. Numerically,
nucleation and condensation are always solved simultaneously, because nucleation and condensation are competing processes.
However, the user can choose to split coagulation from other processes (variable splitting set to 1 in the namelist). To accurately
represent the growth of UFP, the condensation of non-volatile compounds can be solved dynamically for all size sections. The
influence of non-volatile compounds on the growth of UFP was illustrated in a test case in v1.1 and in the guide (Sartelet

et al., 2025a). Semi-volatile inorganic and organic compounds are less likely to condense on UFP, because of the Kelvin effect. Therefore, to gain computational time, the condensation/evaporation of semi-volatile compounds may be solved by assuming bulk thermodynamic equilibrium between the gas and particle phases. An algorithm is then applied to redistribute the bulk aerosol concentrations across the size sections in a thermodynamically consistent way, explicitly accounting for the Kelvin effect, which increases the effective vapor pressure over smaller particles.

## 5.1 Nucleation

Several nucleation parameterizations are implemented. They can be used separately or together. An example of their use in 3D is illustrated in Sartelet et al. (2022). The nucleation parametrizations differ in the gas precursors (sulfuric acid, ammonia, extremely-low volatile organic compounds) and/or in the formulation of the parameterization.

– binary: water and sulfuric acid with the parameterisations of Vehkamaki et al. (2002) or Kuang et al. (2008);

– ternary: water, sulfuric acid and ammonia with the parameterisation of Napari et al. (2002) or Merikanto et al. (2007, 2009). To avoid artificially large nucleation rates in the parameterisation of Napari et al. (2002), a maximum nucleation rate of 1.d6 #particles cm$^{-3}$ is set. A scaling factor of the nucleation rate can be applied (parameter scal_ternary in the namelist).

– heteromolecular: sulfuric acid, and extremely-low volatile organic compounds from monoterpene autoxidation with the parameterisation of Riccobono et al. (2014).

– organics: the user can choose the list of extremely-low volatile organic compounds for which nucleation is taken into account. The nucleation rate can be ajusted by specifying a exponent of the power law (parameter nexp_org) and scaling factor (parameter scal_org):

$$J = scal_{org} \left[ Organics \right]^{nexp_{org}}. \tag{33}$$

Nucleated particles are assigned to the first size bin, whose lower bound should be set to about 1 nm to represent freshly formed clusters. The composition of the nucleation-size bin is determined by the precursor species involved in nucleation, and it is estimated based on the properties of these species.

## 5.2 Growth of ultrafine particles

To illustrate the growth of nucleated nanoparticles, the nucleation test case presented in Sartelet et al. (2006) is simulated. The initial distribution corresponds to the hazy conditions of Seigneur et al. (1986). The sulfuric acid production rate is 0.825 $\mu$g m$^{-3}$ h$^{-1}$, the temperature is 288.15 K and the relative humidity is 60%. The particles are initially assumed to be made of 70% sulfate and 30% ammonium. The initial gas phase ammonia concentration is taken to be 8 $\mu$g m$^{-3}$. The concentrations of gas-phase ammonia and particulate-phase ammonium evolve with time due to both condensation/evaporation and ternary nucleation. The simulation is run for 1 h with output every 60 s. Fig. 16 shows the size distribution of particles at the initial time and after 1 h. The nucleated particles grow under the effect of condensation of sulfuric acid and ammonia and

coagulation. This growth can be accelerated by the presence of extremely low-volatility organic compounds (ELVOCs), such as the SSH-aerosol surrogate "Monomer", underlying the importance of an accurate modelling of the formation of ELVOCs for the growth of ultra-fine particles.

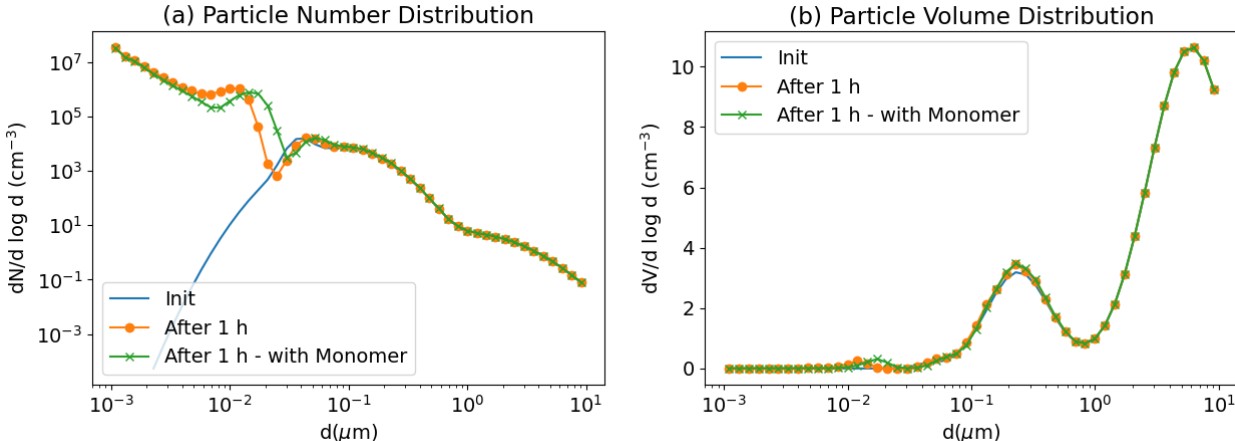

**Figure 16.** Size distribution of particles in the nucleation test case. Under hazy conditions, nanoparticles nucleated from water, sulfuric acid and ammonia grow to larger size. A simulation adding ELVOCs (Monomer) to the initial conditions is performed to illustrate their effect on the growth of ultrafine particles. Number (left panel, a) and volume (right panel, b) concentrations.

## 5.3 Bulk equilibrium and Kelvin effect

To decrease CPU time associated with the treatment of condensation and evaporation, thermodynamic equilibrium between the gas and particle phases can be assumed. In that case, the bulk equilibrium is first computed, and then the bulk aerosol concentrations need to be distributed among the different size sections.

The redistribution was changed in version 2.0 to take into account the Kelvin effect, and for inorganics, the neutralization of acids by positive inorganic ions (sodium and ammonium). The amount of ammonium that condenses to neutralize sulfate

is determined separately from the amount of ammonium that condenses to neutralize other compounds, such as nitric acid. Then the amount of ammonium that neutralizes sulfate is redistributed amongst the size sections, depending on the sulfate concentration of each section. The amount of ammonium that condenses with other semi-volatile compounds such as nitric acid is redistributed amonst the size section using a formulation that depends on the Kelvin effect and the condensation rate (depending on the gas-phase diffusivity of species and the diameters of particles).

To illustrate the role of the Kelvin effect versus the sulfate neutralization, the urban condition test case of Seigneur et al. (1986) is modified by adding $NH_3$ in the initial conditions, and by taking into account only condensation/evaporation processes (nucleation and coagulation are not considered). The initial particle concentration is assumed to be made of sulfate only, and the initial concentration of $NH_3$ is assumed to be 100 $\mu$g m$^{-3}$. The simulation is run for 12 h.

Four simulations are run: in 2 simulations condensation/evaporation is computed dynamically with or without taking into

account the Kelvin effect, in the other 2 simulations thermodynamic equilibrium is assumed. The number and volume size distribution simulated with the four numerical algorithms are compared in Fig. 17. The four numerical algorithms give similar number concentrations, because $NH_3$ condenses to neutralize the sulfate independently of the Kelvin effect.

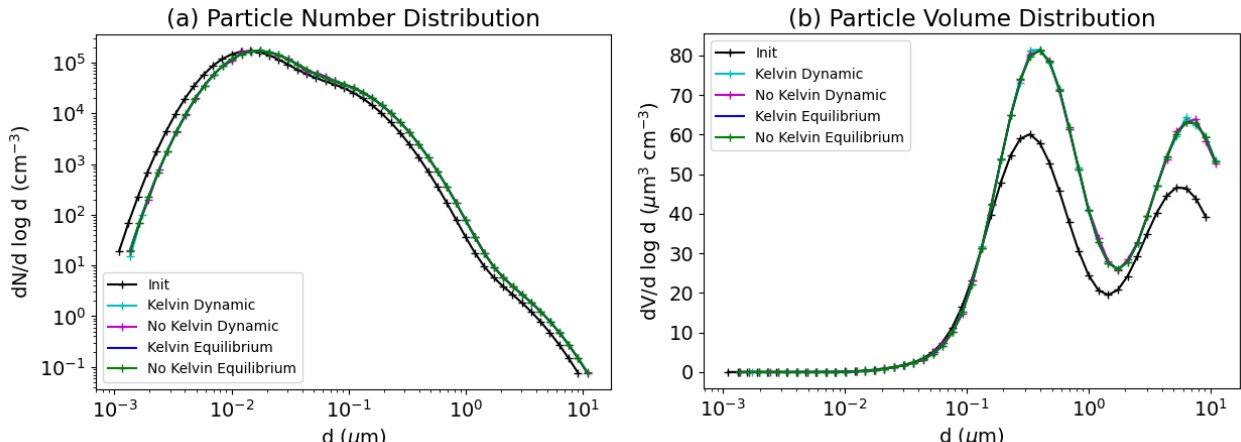

**Figure 17.** Condensation and thermodynamic equilibrium test case for the condensation of ammonia neutralizing sulfate. Number (left panel, a) and volume (right panel, b) concentrations.

In the previous test cases, about 40 $\mu$g m$^{-3}$ of ammonia is used to neutralize sulfate and the remaining concentration stays in the gas phase. To compare the effect of the Kelvin effect between dynamical calculation and thermodynamic assumption, nitric

acid is added in the initial gas-phase concentration. Nitric acid condenses with the remaining ammonia to form ammonium nitrate, as shown in Fig 18. As seen in the comparison of the number size distribution, the particles of diameters below about 0.01 $\mu$m do not grow by the condensation of ammonium nitrate if the Kelvin effect is taken into account, while these particles grow to larger diameters by condensation of ammonium nitrate if the Kelvin effect is not taken into account. The algorithm used to redistribute ammonium nitrate amongst size section is effective at limiting this condensation on particles of small

diameters. This test case is also presented for organic aerosols in the SSH-aerosol guide (Sartelet et al., 2025a).

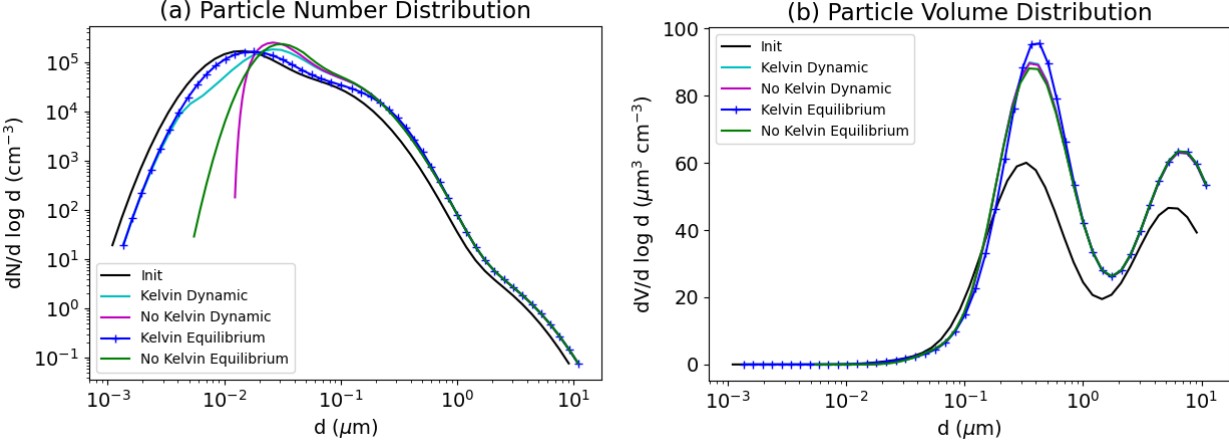

**Figure 18.** Condensation and thermodynamic equilibrium test case for the condensation of ammonia neutralizing sulfate, and for the condensation of ammonium nitrate. Number (left panel, a) and volume (right panel, b) concentrations.

## 6 Coupling to 3D models

SSH-aerosol can be coupled with external (3D) tools using the shared library (libssh-aerosol.so) available after compilation. A prototype of the typical workflow is described in the guide (Sartelet et al., 2025a). A "tool" repertory is added in v2.0. It contains routines that may be useful for the numerical representation of aerosols and the setup of simulations. For example, a routine is defined to partition, i.e. split, a size section into several size sections, while conserving both mass and number concentrations. This may be used to refine the number of size sections for initial, boundary conditions or emissions in 3D-simulations. Also a routine is provided to discretize into sections an aerosol distribution known as the sum of lognormal modes. This routine was used to build some of the test cases, and can be used in 3D simulations to link a lognormal representation of particles to the sectional one of SSH-aerosol.

SSH-aerosol has been coupled with various types of 3D model to simulate air quality. At the regional scale, it was coupled with the transport chemistry models Polyphemus/Polair3D (Sartelet et al., 2022) and CHIMERE (Menut et al., 2024) to study, in particular, the formation of ultrafine particles (Sartelet et al., 2022; Park et al., 2025; Jacquot and Sartelet, 2024) and the influence of organic aerosol patterns of different complexity (Wang et al., 2024; Sartelet et al., 2024). On a local scale, it was coupled to the MUNICH street network model (Kim et al., 2022), as well as to the OpenFoam and Code_Saturne computational fluid dynamic models (Lin et al., 2023), enabling the study of the secondary formation of pollutants in streets, in particular the role of ammonia emissions from vehicles (Lugon et al., 2021; Lin et al., 2023), the formation of condensables (Wang et al., 2023a) and secondary aerosols (Lin et al., 2023, 2024b, a; Sartelet et al., 2024), ultrafine particles (Park et al., 2025). The use of SSH-aerosol in models of different scales has made it possible to model primary and secondary pollutants over a whole city, from the regional down to the street scale (Lugon et al., 2021, 2022; Sarica et al., 2023a; Park et al., 2025; Sartelet et al., 2024;

Squarcioni et al., 2025) and to study at both the regional and local scales the impact of certain processes that are often missing from modelling studies, such as asphalt emissions (Sarica et al., 2023b) or trees (Maison et al., 2024a, b), as well as to assess the impact of scenarios for changes in mobility and traffic (Lugon et al., 2022; Sarica et al., 2024).

One complexity of the use of SSH-aerosol inside 3D air quality models is due to the type of numerical solver used to calculate concentrations, especially if the model does not rely on a splitting approach where aerosol formation is not separated from other 785 processes. However, Couvidat et al. (2025) proposed different methods to couple SSH-aerosol to 3D models depending on the type of numerical solvers.

## 7   Conclusions

The advancements in the SSH-aerosol v2.0 model represent a significant leap forward in the modeling of gas-phase chemistry and aerosol dynamics. By integrating state-of-the-art modules such as SCRAM, SOAP, and $H^2O$, the model offers unparalleled 790 flexibility and precision in simulating the formation and aging of aerosols, describing the properties of particles from the size and mixing-state distribution to the detailed particle composition. New features enhances the model's ability to capture complex atmospheric processes with high fidelity: incorporation of near-explicit chemical schemes, reduced mechanisms generated using GENOA, possibility of defining organic compounds according to their SMILES code or decomposition into functional groups in order to model the partitioning between aqueous and organic phases and the role of non-ideality, particle-phase 795 reactions, composition-dependent viscosity, and fast algorithms to simulate the evolution of ultrafine particles.

The test cases included in the study underscore the robustness and versatility of SSH-aerosol v2.0. Whether modeling the effects of viscosity on gas-particle partitioning, the impact of radical chemistry on SOA formation, the role of NOx levels in influencing secondary organic aerosol yields, or the formation and evolution of ultrafine particles, the model demonstrates its capability to handle a wide range of atmospheric scenarios. The ability to couple with 3D models further extends its appli-800 cability, making it suitable for both localized studies and regional-scale simulations. By bridging the gap between simplified and near-explicit chemical representations, it provides the means to explore and predict the behavior of aerosols under varying environmental conditions with greater accuracy.

Moreover, the improvements in computational efficiency, such as the direct interpretation of chemical reactions, the revision of numerical solvers and faster coagulation algorithms, ensure that the model remains practical for extensive applications in 3D. 805 The addition of a wall-loss module, the modularity of the input data, such as the possibility to setup parameters for nucleation rates, enhance its usability for comparison to chamber experiments.

Overall, SSH-aerosol v2.0 stands out as a comprehensive tool for advancing our understanding of atmospheric chemistry and aerosol dynamics. Future developments could further enhance the model's capabilities, particularly coupling to molecular dynamic model for nucleation.

**Appendix**

The table below lists all the options available in the namelist. Some parameters are not compulsory. If the parameter is not provided, a default value is used. Some parameters are indicated as "Optional", they are not used unless a specific option is activated. In that case, the parameter must be provided. For example, parameter *N_groups* is only used when *tag_external=1*.

| Options | Default value | Description |
|---|---|---|
| Section: setup_meteo | | |
| latitude | Compulsory | Latitude (in degree): use to compute photolysis rate as a function of time and location |
| longitude | Compulsory | Longitude (in degree): use to compute photolysis rate as a function of time and location |
| Temperature | Compulsory | Temperature in K |
| Pressure | Compulsory | Pressure in Pa |
| Humidity | Compulsory | Specific humidity in kg/kg. Either specific or relative humidity must be provided |
| Relative_Humidity | Compulsory | Relative humidity. Either specific or relative humidity must be provided |
| meteo_file | None | File to provide meteorological data as a function of time |
| Section: setup_time | | |
| initial_time | Compulsory | Time (in s) at beginning of simulation |
| final_time | Compulsory | Time (in s) at end of simulation |
| delta_t | Compulsory | General timestep (in s) of SSH-aerosol |
| Section: initial_condition | | |
| with_init_num | 0 | Are initial number concentrations provided? |
| tag_init | 0 | Internally (:0) or externally (:1) mixed aerosols |
| tag_dbd | | Size bound generated (:0) or read (:1) |
| N_sizebin | Compulsory | Number of particle sizebin |
| wet_diam_estimation | 1 | Method used to estimate wet diameter as a function of dry diameter, humidity, and composition. 0: based on Isorropia. 1: based on water concentration. |
| init_gas_conc_file | Compulsory | File on initial concentrations of gases |
| init_aero_conc_mass_file | Compulsory | File on initial mass concentrations of particles |
| init_aero_conc_num_file | Compulsory | File on initial number concentrations of particles |
| Section: initial_diam_distribution | | |
| diam_input | Compulsory | Initial diameter of the size distribution |
| Section: emissions | | |
| tag_emis | 0 | 0: no emissions. 1: internally mixed emissions. 2: externally mixed emissions. |

| with_emis_num | 0 | Put 1 to provide particle number emissions |
|---|---|---|
| emis_gas_file | Optional | Compulsory if *tag_emis* is not set to 0. File with emissions of gases |
| emis_aero_mass_file | Optional | Compulsory if *tag_emis* is not set to 0. File with mass emissions of particles |
| emis_aero_num_file | Optional | Compulsory if *emis_num* is not set to 0. File with number emissions of particles |

| Section: mixing_state | | |
|---|---|---|
| tag_external | 0 | Internally (:0) or externally (:1) mixed aerosols |
| N_groups | Optional | Compulsory if *tag_external=1*. Number of groups of aerosol compounds for which the composition is discretised. |
| N_frac | Optional | Compulsory if *tag_external=1*. Number of mass fraction sections used in the discretisation of composition. |
| kind_composition | 0 | 0: automatic composition discretization. 1: provided by the user. |

| Section: fraction_distribution | | |
|---|---|---|
| frac_input | Optional | Compulsory is *kind_composition=1*. Bounds of mass fraction sections. |

| Section: gas_phase_species | | |
|---|---|---|
| species_list_file | Compulsory | List of gaseous species |

| Section: aerosol_species | | |
|---|---|---|
| aerosol_species_list_file | Compulsory | List of aerosol species. |
| aerosol_structure_file | Optional | File to provide UNIFAC group decomposition of organic aerosol species. |

| Section: physic_gas_chemistry | | |
|---|---|---|
| tag_chem | 0 | Put 1 to activate chemical reactions. |
| attenuation | 1. | value below 1 in case of cloud attenuation of photolysis, and it is equal to 1 if no cloud attenuation (clear sky). |
| option_photolysis | 1 | 1: photolysis rates estimated in the program, 2: read from binary files |
| time_update_photolysis | 100000 | Photolysis rate from binary files read again at this time value in seconds |
| with_heterogeneous | 0 | 1: with heterogeneous reactions, 0: without |
| with_adaptive | 0 | 1: activate automatic timestep to solve chemistry |
| adaptive_time_step_tolerance | 0.001 | Tolerance on relative error accepted to solve chemistry |
| min_adaptive_time_step | 0.001 | Minimal time step (in s) to solve chemistry |
| RO2_list_file | Optional | List of $RO_2$ radical species accounted for in the $RO_2$ pool. |
| tag_RO2 | 0 | 0: no reaction with $RO_2$ pool, 1: only generated $RO_2$, 2 only background $RO_2$, 3: both background and generated $RO_2$ |

| | | |
|---|---|---|
| photolysis_dir | "./photolysis/" | Directory where binary files on photolysis data are. |
| photolysis_file | "./photolysis/ photolysis- cb05.dat" | List of Photolysis reactions |
| n_time_angle | 9 | Number of time angles in photolysis binary files |
| nsza | 11 | Number of solar angles used in reactions file when option_photolysis = 1 |
| time_angle_min | 0 | Minimal time angle in binary files (in hour) |
| delta_time_angle | 1 | Resolution of time angle in binary files (in hour) |
| n_latitude | 10 | Number of latitude levels in photolysis binary files |
| latitude_min | 0 | Minimal latitude in binary files |
| delta_latitude | 10 | Resolution of photolysis binary files (in degree) |
| n_altitude | 9 | Number of altitude levels in photolysis binary files |
| altitude_photolysis_input | 0.0, 1000.0, 2000.0, 3000.0, 4000.0, 5000.0, 10000.0, 15000.0, 20000.0 | Altitude levels in photolysis binary files (in meter) |
| keep_gp | Advanced: 0 | If 1, the gas/particle partitioning of SVOC compounds is assumed to be constant. To be used only in the equilibrium mode with high time step. Expert parameter. Should be used with caution. |
| kwall_gas | 0. | Wall loss rate of gaseous SVOC (in $s^{-1}$). Fixed value. |
| kwall_particle | 0 | Wall loss rate of particles (in $s^{-1}$). Fixed value. |
| Cwall | 0 | wall equivalent concentration (in $\mu g/m^3$). Has to be provided to account for the wall losses of SVOC. |
| eddy_turbulence | | eddy diffusion coefficient (in $s^{-1}$). Used to compute *kwall_gas* and *kwall_particle* according to chamber characteristics (and also diameters for kwall_particle). |
| surface_volume_ratio | 0 | Surface on volume ratio of the chamber (in $m^{-1}$). Used to compute *kwall_gas* and *kwall_particle* according to chamber characteristics |
| kwp0 | 0 | minimal wall loss rate of particles due to electrostatic forces. Used to compute *kwall_particle* as a function of diameters. |
| radius_chamber | 0 | Radius of the chamber (in m). Used to compute *kwall_particle* according to chamber characteristics. |
| | | |
| Section: physic_particle_numerical_issues | | |
| DTAEROMIN | 1.e-5 | Time step used to solve aerosol formation and evolution (in s) |

| redistribution_method | 0 | 0: no redistribution, 3: euler_mass, 4: euler_number, 5: hemen, 6: euler_coupled, 10: Moving Diameter, 11: SIREAM, 12: euler_couple_siream |
|---|---|---|
| with_fixed_density | 1 | 1: Constant density. 0: density computed as a function of composition |
| fixed_density | $1.4 \times 10^3$ | Default value for aerosol density (in kg/m$^3$) |
| splitting | 1 | 0: coagulation and (condensation/evaporation+nucleation) are splitted (solved separately). 1: the processes are solved together. |

**Section: physic_coagulation**

| with_coag | 0 | With coagulation (1) or not (0) |
|---|---|---|
| i_compute_repart | 1 | 0: repartition coefficient are read. 1: ,repartition coefficient are computed |
| i_write_repart | 0 | Put 1 to write repartition coefficient after calculation. |
| Coefficient_file | Optional | Path to (written or read) repartition coefficient |
| Nmc | $10^6$ | Number of Monte Carlo points used to compute repartition coefficients |

**Section: physic_condensation**

| with_cond | 0 | With condensation (1) or not (0) |
|---|---|---|
| Cut_dim | Compulsory | Diameter (in $\mu$m) above which the dynamics condensation/evaporation of inorganic (and also organics for *soap_inorg=1*) is solved |
| ISOAPDYN | 0 | 0: partitioning of SVOC at equilibrium, 1: solved dynamics of SVOC condensation/evaporation |
| IMETHOD | 1 | numerical solver used to solved the dynamics of condensation/evaporation of SVOC. 0: explicit method. 1: implicit backward Euler. |
| soap_inorg | 0 | Thermodynamic module for inorganics. 0: ISORROPIA, 1: SOAP (coupled thermodynamics between organics and inorganics) |
| nlayer | 1 | Number of layers in the organic phase of aerosols. |
| with_kelvin_effect | 1 | 1: with kelvin_effect, 0: without |
| tequilibrium | 0.1 | characteristic time above which condensation/evaporation of SVOC is solved dynamically (for ISOAPDYN=1 and IMETHOD=0) |
| dorg | $10^{-12}$ | Diffusion coefficient in the organic phase in m$^2$/s. Used for *nlayer>1*. If set to 0, the diffusion coefficient is computed as a function of composition |
| coupled_phases | 0 | Parameter used to track (1) or not (0) concentrations in the aqueous phase separateley from the organic phase. Have to be set to 1 when some compounds are both hydrophilic and hydrophobic and when *ISOAPDYN=1*. |
| activity_model | 3 | Option to compute activity coefficients of organic compounds. 1: ideal, 2: UNIFAC (short range interactions), 3: AIOMFAC (short, medium and long range interactions) |

| | | |
|---|---|---|
| epser | 0.01 | Tolerance for the relative error for time step adjustment on the condensation/evaporation of inorganic compounds |
| epser_soap | 0.01 | Tolerance for the relative error for time step adjustment on the condensation/evaporation of organic compounds |
| niter_eqconc | Advanced: 1 | Recompute local equilibrium with the thermodynamic module every niter_eqconc iterations. Can be used in coupling with 3D models to decrease CPU time. Expert parameter. Should be used with caution. |
| niter_water | Advanced: 1 | Recompute water concentrations with the thermodynamic module every niter_eqconc iterations. Can be used in coupling with 3D models to decrease CPU time. Expert parameter. Should be used with caution. |
| co2_conc_ppm | 410 | $CO_2$ concentrations (in ppm). Used only when carbonates are simulated with *soap_inorg=1*. |
| NACL_IN_THERMODYNAMICS | 0 | 0: Consider $Na^+$ and $Cl^-$ in the thermodynamic modules. |
| SOAPlog | 2 | Parameter to extract Henry's law parameter (calculated from the structure and the saturation vapor pressure) and the smiles decomposition in UNIFAC functional groups calculated by the model. 0=no output, 1=on screen, 2=in written files (*smile2UNIFAC.decomp* and *henryfromSOAP.dat*). |
| reaction_soap_file | No reactions | Path to a file listing all intra-particle reactions. *Example given by species-list/REACTIONS.particle.soap* |

Section: physic_nucleation

| | | |
|---|---|---|
| with_nucl | 0 | With nucleation (1) or not (0) |
| nucl_model_binary | 0 | type of binary nucleation model considering water and sulfuric acid. 0=none, 1=Vehkamaki et al. (2002), 2=Kuang et al. (2008) |
| nucl_model_ternary | 0 | type of ternary nucleation model considering water, sulfuric acid, and ammonia. 0=none, 1=Napari et al. (2002), 2=Merikanto et al. (2007, 2009) |
| scal_ternary | 1 | Scaling factor for ternary nucleation rate. Ternary nucleation rate with be multiplied by the scaling factor. |
| nucl_model_hetero | 0 | With (1) or without (0) heteromolecular nucleation. |
| scal_hetero | 0.1 | Scaling factor for heteromolecular nucleation rate. Heteromolecular nucleation rate with be multiplied by the scaling factor. |
| nesp_org_h2so4_nucl | 0 | Number of organics involved in heteromolecular nucleation |
| name_org_h2so4_nucl_species | Optional | List of organics species involved in heteromolecular nucleation. Must have the same dimension than *nesp_org_h2so4_nucl*. |
| nucl_model_org | 0 | With (1) or without (0) organic nucleation. |
| scal_org | 0.1 | Scaling factor for organic nucleation. Nucleation rate with be multiplied by the scaling factor. |

| | | |
|---|---|---|
| nexp_org | 0.1 | Exponent of the power law in the computation of the organic nucleation rate. |
| nesp_org_nucl | 0 | Number of organics involved in organic nucleation |
| name_org_nucl_species | Optional | List of organic species involved in nucleation. Must have the same dimension than *nesp_org_nucl*. |
| Section: output | | |
| output_directory | Compulsory | Path to the output dir. |
| output_type | 1 | 0: no output, 1: text files, 2: binary files, 3: NETCDF files. |
| particles_composition_file | Compulsory | Discretisation of the particle composition determined by the model in case mixing-state is followed. The file is made into output_directory. |
| output_aero_list | All species | List of aerosol species in output. Used when *output_type=0* |
| output_gas_list | All species | List of gaseous species in output. Used when *output_type=0* |

*Code availability.*  The code is available in the github platform (https://github.com/sshaerosol/ssh-aerosol),
and in https://doi.org/10.5281/zenodo.14196277 (Sartelet et al., 2025b).

*Author contributions.*  Conceptualization: KS, FC; formal analysis: KS, FC, ZW, VL, YK; funding acquisition: KS, FC; investigation: KS, FC, ZW, VL, YK; methodology: KS, FC, ZW, VL, YK; project administration: KS, FC; software: KS, FC, ZW, VL, YK; validation: KS, FC, ZW, VL, YK; visualization: KS, FC, ZW, VL, YK; writing – original draft preparation: KS, FC, ZW, VL, YK; writing – review & editing:
KS, FC, ZW, VL, YK.

*Competing interests.*  The authors have no competing interests to declare.

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
