# Peer review of "Advanced modeling of gas chemistry and aerosol dynamics with SSH-aerosol v2.0"

_EGUsphere, 2025_

## Referee Comment (RC2)

**Review of Advanced modelling of gas chemistry and aerosol dynamics with SSH-aerosol v 2.0**

This manuscript describes the development and updates of the model SSH-aerosol, which can be utilized to simulate the formation of secondary aerosols, and evolution of both the primary and secondary aerosols. This involves simulating the gas-and particle phase heterogenous chemistry and aerosol dynamics. Various test cases ranging from aerosol dynamic processes and gas-particle phase chemistry were simulated and presented in the manuscript to show model capabilities. The manuscript is well written, with the supplement/guide serving as a detailed blueprint for potential users to test an implement the model. The SSH-aerosol model serves as an interesting tool to simulate aerosol dynamics under different environmental scenarios and would surely benefit the modelling and experimental community. I recommend the publication of this manuscript, after the authors have addressed the following comments/suggestions.

**General comments:**

I would recommend adding a figure outlining the different inputs and modules and their workflow. This could also serve the purpose of being a visual reference for any user trying the model.

Add figure letters (e.g. Figure 4, a, b or c) to avoid confusion.

While discussing results (e.g. L 303-306, Figure 8 etc.), the authors states what the results shows, but don't discuss why that's happening. The readers will appreciate if they get to know for e.g., why SOA yields are increasing/decreasing in the presence of NOx.

Nucleation: This section has not been discussed in depth but rather just glossed over.

Model computational time: How much impact does different configurations (gas+SOA schemes) have on the computational time. Since this model can be couple to a 3-d model, how realistic is it to use particle phase and viscosity calculation with the coupled 3-d model since its includes discretised particle layer calculations.

**Minor comments:**

L 35: "atmospheric chambers"- I believe the correct terminology would be atmospheric smog chambers.

L 43: "Several aerosol box models exist". I think this line is unnecessary. Either remove it or club it with the next line.

L56-57: One must be careful in formulating such descriptions. MCM is an explicit gas phase oxidation mechanism, not a SOA formation scheme. It is coupled to a SOA formation scheme and therefore should be categorized as a chemical scheme and not a SOA scheme.

L65-66: Some examples of different reduction strategies could be included.

L89: "repertories" to repositories

L 99: what kind of solver is used to deal with the ODE systems? How will it affect the computational time?

L 110: How are the saturation vapor pressures generated? Nannolal/Evaporation/SIMPOL or another mechanism?

L211 – 213: does this mean that one can use 2 or more chemistry schemes at the same time? For example, one can use CB05 for estimating a reaction products conc., say X and another explicit scheme to use the [X] in their respective chemical schemes? If so, does it only work for first generation oxidation products or more complex oxidation products as well? Are these conc. profiles interpolated to the timestep of the SOA chemical scheme?

L214-216: This part is unclear. Does it mean one can have a specific combination for each precursor + SOA scheme? E.g. alpha-pinene using MCM + beta caryophyllene using RACM2? if so how would one account for example RO2 from ap +RO2 (bcarp) reaction products from different chemical schemes, in case the resulting RO2 is not present in either scheme?

L223-224: There are different vapor pressure estimation methods on the UManSysProp. Which method was used?

L234: is this analogous to "lumped RO2" species? Is this RO2 pool rereferring to RO2 from one precursor alone or can it refer to lumped RO2 from all of MCM or a subset of precursors?

L 236: what does "ARR" stand for?

L 243: "TBRO2" is not referenced in the equation above or explained anywhere else.

Figure 2: This makes sense since MCM doesn't yet include a fully developed peroxy radical autooxidation scheme for beta caryophylle. It would be interesting to perform such tests with apha-pinene.

Section 3.2 My one suggestion here to improve readability would be to have a table detailing the explicit and reduced mechanism, with a shorthand representing each. For e.g near explicit mech for monoterpenes could be denoted by  $\text{Exp}_{\text{mcm+pram}}$  or something like that. It could make it easier for readability without one having to go back and forth to see what the respective mechanisms were.

L281: This should be right panel.

L 294: upper right panel?

L296-294: Referring to Fig 4: these differences are barely visible in the plot. Perhaps the authors will think about providing percentage increase or decrease w.r.t the reference to indicate the difference.

L 299-303: why is that?

Figure 3: what is the reference scheme? it should be mentioned in the caption. Is the related to NO2 conc.?

L 364-365: Since H2O is based on smog chamber experiments, was wall losses considered when simulating SOA for species which was compared to H2O? I suspect the SOA yields for species where H2O was used for comparison would differ if wall losses of organics would be considered.

L 378-380: why does the H2O mechanism overestimate the SOA mass?

L398: "interface the index of the layer at the interface". what does this represent or mean? Also  $K_{,bin}$ , interface,  $p_{,i}$  and the corresponding  $M_o^{bin,interface \, layer}$  is not defined.

L 413: "f". This needs more explanation here. For e.g., what does a value f=0.2 indicate? how is f determined?

L 430: "Compounds not affected to a specific molecular structure". what does this mean?

Figure 9: I assume that since this is an organic phase there would be no charged compounds would be present in the particle phase. But when the RH increases compounds in the particle phase are bound to break and form cations and anions. How will this affect the viscosity especially in the presence of acids?

Figure 9: Add panel letters to make it easy to follow. how does the c panel which is bottom left change if you have varying RH for alcohol+acid + varying RH?

L480: parameter k. Is this a function of water conc. and pH?

L 505: Kmaxoligo, hoe is this determined?

L505: amonomer, I would suggest representing this as Eta or sum(aA, monomer), both in the equation 24 and here.

Eq. 28: is this nomenclature based on H2O or MCM or other chemical schemes?

L 587: SOAP. Does the model account for the formation of salts in the particle-phase?

Figure 13: please improve the legend. It's hard to read it in its current form.

Nucleation: Have the authors considered to couple ACDC for more complex termolecular acids and bases?

---

## Author Comment (AC1)

**Advanced modeling of gas chemistry and aerosol dynamics with SSH-aerosol v2.0**

Karine Sartelet[1], Zhizhao Wang[1,2,3], Youngseob Kim[1], Victor Lannuque[2], and Florian Couvidat[2]

[1]CEREA, Ecole nationale des ponts et chaussées, EDF R & D, Institut Polytechnique de Paris, IPSL, 77455 Marne-la-vallée, France
[2]INERIS, 60550 Verneuil en Halatte, France
[3]Now at University of California, Riverside, CA, USA, and National Center for Atmospheric Research, CO, USA

**Correspondence:** Karine Sartelet (karine.sartelet@enpc.fr), Florian Couvidat (florian.couvidat@ineris.fr)

**Reply to reviewer 1**

*The paper by Sartelet and co-workers describes the new developments in the established aerosol model "SSH" (Sartelet et al., 2020), now termed SSH-2.0. The additions and changes to the model are comprehensive, which justifies publication of this update. Code is provided on github and Zenodo. One of the biggest advantages of this very complete aerosol model is its capability to be run both as box model to describe laboratory experiments, and as module in larger scale models. The authors added functionalities that seem helpful for both tasks. The authors have a difficult task at hand with this paper to describe the full capability of the model. This work details the development of advancements in gas-particle partitioning, viscosity and particle-phase reactions into the model, which I think is a good choice. However, I think the paper would benefit from some general clarifications about the SSH model and how it works (even though this has been established in previous work), possibly aided by some diagrams. After the questions and comments below are addressed, this will be a nice paper that fits well within the scope of GMD.*

**General Comments**

*Some of the new model capabilities advertised in Abstract and Conclusions are not sufficiently discussed in the paper, especially the treatment of new particle formation / nucleation and ultrafine particles. It would be good if these points would be strengthened.*

**Our reply**: To strengthen the section on ultrafine particles, more details about the treatment of coagulation have been added at the beginning of section 5.

"For coagulation, the discretized equations governing aerosol dynamics involve partition coefficients that account for the possibility that the coagulation of particles from two given size sections may produce particle sizes spanning multiple sections. The size mesh is assumed to remain fixed over time, so that these partition coefficients can be pre-computed. In previous model

version, the coagulation partition coefficients were calculated using a Monte-Carlo approach. ..... Because the explicit formulation has negligible computational cost compared with the Monte Carlo approach; only the explicit formulation is retained for internally-mixed particles in v2.0."

25 More details on the size distribution are also added:

"Given the strong impact of coagulation on the growth of ultrafine particles, it should always be included in their modelling. Consequently, the particle-diameter limits are held constant over the course of the simulations. Because nucleation corresponds to the formation of nanometric particles from gas precursors, the lowest particle diameter bound of the size distribution should be about 1 nm."

30 In the nucleation section, the following explanations are also added: "The nucleation parametrizations differ in the gas precursors (sulfuric acid, ammonia, extremely-low volatile organic compounds) and/or in the formulation of the parameterization. ", as well as the formula for organic nucleation:

$$J = scal_{org} \left[Organics\right]^{nexp_{org}}. \tag{1}$$

Nucleated particles are assigned to the first size bin, whose lower bound should be set to about 1 nm to represent freshly formed
35 clusters. The composition of the nucleation-size bin is determined by the precursor species involved in nucleation, and it is estimated based on the properties of these precursors."

A section describing the growth of ultrafine particles has also been added. "To illustrate the growth of nucleated nanoparticles, the nucleation test case presented in Sartelet et al. (2006) is simulated. The initial distribution corresponds to the hazy
40 conditions of Seigneur et al. (1986). The sulfuric acid production rate is 0.825 $\mu$g m$^{-3}$ h$^{-1}$, the temperature is 288.15 K and the relative humidity is 60%. The particles are initially assumed to be made of 70% sulfate and 30% ammonium. The initial gas phase ammonia concentration is taken to be 8 $\mu$g m$^{-3}$. The concentrations of gas-phase ammonia and particulate-phase ammonium evolve with time due to both condensation/evaporation and ternary nucleation. The simulation is run for 1 h with output every 60 s. Fig. 1 shows the size distribution of particles at the initial time and after 1 h. The nucleated particles grow
45 under the effect of condensation of sulfuric acid and ammonia and coagulation. This growth can be accelerated by the presence of extremely low-volatility organic compounds (ELVOCs), such as the SSH-aerosol surrogate "Monomer", underlying the importance of an accurate modelling of the formation of ELVOCs for the growth of ultra-fine particles. "

[Figure]

**Figure 1.** Size distribution of particles in the nucleation test case. Under hazy conditions, nanoparticles nucleated from water, sulfuric acid and ammonia grow to larger size. A simulation adding ELVOCs (Monomer) to the initial conditions is performed to illustrate their effect on the growth of ultrafine particles. Number (left panel, a) and volume (right panel, b) concentrations.

*Something that would be really instructive in this study would be to give the reader an idea how much more time consuming the model gets when the complexity is increased, e.g. by including more complex chemistry (e.g., Figs. 2, 3 etc.) or including particle-phase effects (e.g., Fig. 8). Could you time the model and report the runtime?*

**Our reply:** The runtime is now compared in the table 1 and discussed in the paper.The following lines are added in section 2.5, dedicated to the runtime.

As detailed in the appendix, many options are available, allowing the representation of the different physical processes with a large range of complexity. It is illustrated with different test cases in this paper. Increasing complexity may often lead to a large increase in computational time (CPU), as illustrated in Table 1. Near-explicit chemical schemes may be 3 to 65 times more expensive than parameterized schemes (comparisons of simulation numbers 1, 2,3; numbers 4 and 7). Taking into account ideality has a limited impact on CPU time, with an increase by a factor about 1.5-2 (comparisons of simulation numbers 4 and 5; numbers 4 and 6; numbers 9 and 10). The impact of oligomerisation on CPU time may be limited (comparison of simulation numbers 12 and 13), while the impact of viscosity may be high, with an increase by a factor almost 8 (comparison of simulation numbers 10 and 11). Dynamic gas/particle partitioning may lead to a large increase of CPU time, from a factor 2 to a factor 20 (comparisons of simulation numbers 13 and 14; numbers 15 and 16; numbers 17 and 18; numbers 19 and 20). The relations between CPU time and model complexity presented here are only illustrative, and they may vary depending on the environmental conditions specified in the model's input.

**Table 1.** Runtime (in s) for different simulations presented in the Figures of this paper. Rdc. stands for reduced and Expl. for quasi-explicit.

| Simulation number | Simulation name | Figures | Runtime (s) |
|---|---|---|---|
| 1 | Rdc. SOA scheme | 4, 5 and 6 | 20 |
| 2 | Expl. SOA scheme | 4, 5 and 6 | 65 |
| 3 | $H^2O$ SOA scheme | 4, 5 and 6 | 1 |
| 4 | Expl. toluene SOA scheme - Ideal | 8a | 6 |
| 5 | Expl. toluene SOA scheme - Unifac | 8a | 10 |
| 6 | Expl. toluene SOA scheme - Aiomfac | 8a | 11 |
| 7 | Rdc. toluene SOA scheme - SART24-1 | 8b | 2 |
| 8 | Rdc. toluene SOA scheme - SART24-2 | 8b | 1 |
| 9 | Naphthalene SOA scheme - Ideal | 9 | 15 |
| 10 | Naphthalene SOA scheme - Aiomfac | 9 | 23 |
| 11 | Naphthalene SOA scheme - Aiomfac-visc | 9 | 162 |
| 12 | Isoprene SOA - Without oligomerization | 11 | 2 |
| 13 | Isoprene SOA - With oligomerization | 11 | 3 |
| 13 | $CaCO_3$ equilibrium no $HNO_3$ | 14 | 3 |
| 14 | $CaCO_3$ dynamic no $HNO_3$ | 14 | 28 |
| 15 | $CaCO_3$ equilibrium with $HNO_3$ | 14 | 2 |
| 16 | $CaCO_3$ dynamic with $HNO_3$ | 14 | 43 |
| 17 | Kelvin dynamic | 15 | 0.4 |
| 18 | Kelvin equilibrium | 15 | 0.2 |
| 19 | Kelvin dynamic | 16 | 2.7 |
| 20 | Kelvin equilibrium | 16 | 0.2 |

*I think it would be generally worthwhile to add a short, general section on the numerical implementation of the model code*
65 *(also to prepare Sect. 4.1, which details some of the improvements in SSH-2.0):*
**Our reply:** Section 2 (model structure and new features) is now divided into 4 subsections: Model structure, Physical processes, New features and a new subsection on Numerical implementation, which has been added.

*- I believe this is FORTRAN code?*
70 **Our reply:** The code is written in FORTRAN, except for SOAP which is written in C++. This is now specified at the beginning of section "Model structure and new features".

*- Which ODE solvers are being used?*

**Our reply:** Different ODE solvers are used depending on the processes, this is now detailed in the "Numerical Implementation" section.

*- How is numerical convergence assured? Can the user change numerical integration tolerances?*

**Our reply:** For each process, the initial time step is set to the minimum value specified by the user, and it is subsequently adjusted adaptively according to the user-defined numerical tolerance. This is now specified in the "Numerical Implementation" section.

*- Are there problems with numerical stiffness, if yes, how can they be dealt with? For example, can evaporation lead to the full vanishing of layers?*

**Our reply:** Stiffness is handled using dedicated stiff solvers, as now detailed in the Numerical Implementation section. Evaporation can indeed lead to the complete depletion of certain size bins. To ensure numerical robustness in such cases, the model employs mass-threshold parameters (TINYM and TINYN) that control accuracy and prevent the accumulation of non-physical residual masses. For layers, the volume ratio of layers is assumed to remain constant. Therefore, a layer cannot vanish.

*For many of the examples, only the primary model output (e.g., SOA mass) is provided. I think this paper would be showing the capabilities of SSH-2.0 better if you would also show other model output, such as the concentrations of certain product or intermediates, particle size distributions, or diffusion gradients inside the particles.* **Our reply:** In the manuscript, we present different types of model outputs depending on the purpose of each test case. While Figures 1–8 focus on SOA mass, providing detailed speciation for these cases would require an extensive discussion of the chemical schemes and intermediates, which would detract from the clarity of the comparison. However, several other figures already illustrate the additional outputs suggested by the reviewer.

– Chemical composition and intermediates: Figure 9 explicitly demonstrates the role of composition through its impact on viscosity. Figure 12 presents the formation of organosulfates and methyltetrols alongside SOA mass, and Figure 13 further details particle chemical composition.

– Particle size distributions: Size-resolved outputs are shown in multiple test cases, including Figures 14, 15, and 16, which illustrate the evolution of ultrafine particle size distributions.

In summary, while SOA mass is highlighted in some examples for clarity, the manuscript already includes several cases where product concentrations, intermediates, and size distributions are analyzed to demonstrate the broader capabilities of SSH-aerosol v2.0.

*It does not become clear enough how far the model can be customized by defining one's own particle-phase chemistry, or if only the four types of particle-phase chemistry that is described in section 4.3 and 4.4 can be used.*

**Our reply:** The model can be customized by implementing user-defined particle-phase chemistry, provided it is expressed

using the five reaction types described in Section 4.3. Additional reaction types can in principle be accommodated, but their inclusion would require further code development.

*The English language could be improved at times (especially singular and plural nouns - for example, "dynamics" and "physics" are used as plural nouns in the English language), but language copy-editing will take care of that. Overall, the methods in the paper are a bit difficult to understand at times or remain unexplained. I tried to outline the concepts which I find difficult in the specific comments below.*

**115 Specific Minor Comments**

*l. 47 - The sectional approach is not well described in this paper (despite being mentioned in the abstract). Is the approach to size distributions in this model using fixed or moving bins? This is important for discussing the implementation of nucleation in Sect. 5.1 later. In l. 342, I think size bins are mentioned for the first time without prior introduction. It would be good to clarify how size distributions work in the model.*

120 **Our reply:** To clarify the size distribution, the following paragraph was added at the beginning of section 2: "The size distribution is represented using a sectional approach (Debry et al., 2007), in which the particle population is approximated by discrete size sections characterized by their mass, number, and mean diameter. Depending on user settings, the section boundaries may either remain fixed or evolve in time, and the mean diameter may be constrained to the geometric midpoint of the section or allowed to vary within its bounds. Particles are assumed to be spherical, so that the mass $M$, number $N$, and mean diameter $d_m$

125 in each section satisfy $M = N\rho\pi/6d_m^3$, with $\rho$ the particle density, which can be fixed or vary with the particle composition."

*l. 68 "oftheir" -> "of their"*

**Our reply:** Modified

*l. 106 What is this "collision factor"?*

130 **Our reply:** In the coagulation formulation, the collision factor represents the probability that a particle–particle encounter leads to effective coalescence. It multiplies the collision kernel to give the actual coagulation rate. This is added in the text.

*l. 162 "isoprene SOAs prefer to condense onto aqueous phase" - I believe this is not clear. With "SOAs" you mean oxidation products here?*

135 **Our reply:** "In this paper, the term SOA is used in a broad sense to refer to the compounds involved in the formation of organic particles, regardless of whether they are in the gas or particle phase." This is now added in the introduction. Furthermore "isoprene SOAs" is replaced by "isoprene oxidation products".

*l. 211-224: This paragraph is hard to read for people not deeply familiar with these reaction schemes. What I find most 140 confusing is the use of "oxidant" in quotation marks, describing the schemes (?), alongside oxidant without quotation marks. Why are they called "oxidant" schemes?*

**Our reply:** These "oxidant" schemes correspond to chemical schemes describing the formation of oxidants. For simplicity, this term has been removed, and the paragraph rewritten: "The inorganic reactions governing oxidant formation do not always need to be included in the reaction list. Depending on the user's settings, oxidant concentrations can either be prescribed using constant input profiles, or they can be computed using existing gas-phase chemistry mechanism, such as CB05, RACM2 or MELCHIOR2. These schemes can be complemented with SOA chemical schemes describing the degradation of VOCs leading to SOA formation. To offer flexibility, the model allows users to construct customized chemical mechanisms, covering both gas-phase and SOA chemistry, by combining a chosen gas-phase mechanism with one or more SOA schemes (typically one per SOA precursor class). For VOCs already represented explicitly in the selected gas-phase mechanisms (e.g. toluene, monoterpenes), the associated SOA scheme can be added without modifying oxidant concentrations. However, some VOCs are not included in these gas-phase mechanisms, in which case the inclusion of specific reactions for radicals, oxidants as well as SOA formation may be necessary. For example, the impact of naphthalene on ozone and radical production is not represented in the CB05, RACM2 or MELCHIOR2 mechanisms."

*l. 254 "The influence of the RO2 pool option is low for BCARY." - Why is this the case?*

**Our reply:** The $RO_2 + RO_2$ pathways, i.e., the part influenced by the background $RO_2$ pool, is not the dominant pathway for BCARY SOA formation. This has been added in the paper.

*Figure 2 - "SOA yield" is not defined in the manuscript.*

**Our reply:** Its definition has been added (the fraction of th BCARY's reacted mass that is converted into SOA).

*Figure 3 - Please explain "Rdc" and "Expl." somewhere.*

**Our reply:** Added.

*l. 282 "This is partly attributed to the impact of molecular rearrangement present in the near-explicit scheme, but which is missing in the H2O scheme." - This information does not help the reader understand what is going on. I would suggest to explain it briefly so it can be understood, or only refer to the original study.*

**Our reply:** Explanation has been added: "with ring opening of a bicyclic peroxy radical (BPR) with an O-O bridge (Iyer et al., 2023)"

*l. 294 - "(upper left panel of Fig4)" - I believe this should read "(upper right panel of Fig. 4)". Because you explain first the upper right panel in the figure caption, I believe the order was reversed at some stage in the paper writing process. I think it would be good to discuss the figures from left to right.*

**Our reply:** The figures are indeed discussed from left to right, and thank you for pointing out the mistake in the labeling.

*Figure 6 - Since this chapter is about wall losses: what is the effect of the gas and particle wall loss parameterizations here?*

**Our reply:** This chapter does not focus on wall losses; rather, it presents comparisons with chamber experiments. Wall-loss processes are briefly described at the beginning of the section solely to document the implementation used in the simulations. The core of the section examines SOA formation from various precursors against chamber data, and Fig. 6 specifically shows the results for $\alpha$-pinene.

*l. 364 - What is "radical equilibrium"?*

**Our reply:** This has been removed.

*l. 381–385 - Figure 8 is hardly discussed in the paper. Figure 8 shows a better model-experiment agreement when using AIOMFAC-visc. Can this be understood? Is Naphthalene SOA expected to be more viscous than toluene SOA (Fig. 7)? What would the effect of using AIOMFAC-visc be on the calculations shown in Fig. 7, does it also affect this calculation?*

**Our reply:** For naphthalene SOA, the agreement with experiments is marginally improved when using AIOMFAC-visc compared with AIOMFAC. When accounting for viscosity, diffusion inside the particles limits the condensation of semivolatile species because the particles are semisolid. However, both simulations remain very close to the experimental data, one slightly overestimating and the other slightly underestimating, so this difference is not considered meaningful in regard of uncertainties on simulation parameters (such as the initial diameter of particles). For toluene SOA, viscosity has only a minor influence, likely because their oxidation products are less multifunctional. These sentences have been added to the paper.

*l. 394 - "As in Couvidat and Sartelet (2015) and in SSH-aerosol v1.1, it does not represent the exchange of compounds between layers but assumes a characteristic time to reach the interface." - Can the authors comment why this implicit layer-to-interface approach has been chosen over a more explicit layer-to-layer diffusion approach? Do I understand correctly that the diffusion to the interface is only determined by the composition of the origin layer, not the composition of the layers between this layer and the interface? If yes, the authors should comment on the potential shortcomings of this method. A schematic figure might help the reader to visualize how diffusion is treated in the model.*

**Our reply:** An explicit method is implemented within SSH-aerosol but is very CPU time comsuming, especially when coupled to activity coefficients and a SOA mechanism containing several hundreds of semivolatile organic species. Using an explicit representation would have resulted in a prohibitive CPU time and parallelization within SSH-aerosol would be necessary.

Some details and an illustration were added to the text: "As illustrated by Fig. 2, diffusion is represented by exchange fluxes between the different layers inside the particle and the interface layer. The main consequence of this simplification is that the diffusion of a compound is calculated as if the compound has the same affinity with every layers. It may therefore miss some entrapment effect (impossibility for compounds to cross a layer because of a lack of affinity)."

*Equation 10 - The use of long words in italic font as indices make the equations quite hard to read. Generally it is customary to use italic letters for variables representing numbers, but upright characters for words that do not ("diff"). It is not clear what bin, p and o stand for. What is the unit of J? Typically, flux is expressed as molecules per time and area (e.g., Pöschl, Rudich,*

[Figure]

**Figure 2.** Illustration of implicit treatment of diffusion inside a particle.

*Ammann 2007). Is this the case here (and e.g. in Eq. 23)?*

**Our reply:** The text and the equation have been revised to better explain the unit and remove the italic font for words. We kept the notation of Couvidat and Sartelet (2015) where $A_p$ design the particle phase concentration and $M_o$ the organic aerosol mass. The text now states: The flux of diffusion for a specific layer, bin and species ($J_{\text{diff},i}^{\text{bin,layer}}$ in $\mu$g m$^{-3}$ of air s$^{-1}$) is calculated as:

$$\quad J_{\text{diff},i}^{\text{bin,layer}} = k_{\text{diffusion}}^{\text{bin,layer}} \left( A_{p,i}^{\text{interface,layer}} \frac{K_{p,i}^{\text{bin,layer}} M_o^{\text{bin,layer}}}{K_{p,i}^{\text{bin,interface}} M_o^{\text{bin,interface}}} - A_{p,i}^{\text{bin,layer}} \right) \tag{2}$$

with $k_{\text{diffusion}}^{\text{bin,layer}}$ the parameter of diffusion rate from the interface to the layer (in s$^{-1}$), "interface" the index of the layer at the gas/particle interface, $K_{p,i}^{bin,layer}$ (in m$^3$ $\mu$g$^{-1}$)the partitioning constant, $M_o^{bin,layer}$ (in $\mu$g m$^{-3}$ of air) the mass of the organic phase in the layer and $A_{p,i}^{bin,layer}$ the particle concentrations of the species i in the layer (in $\mu$g m$^{-3}$ of air).

220      *l. 413 "By $B\hat{b}in,layer\_p,i(t+\Delta t)$ is the concentration estimated with Eq. 11" - Do you mean that the result of Eq. 11 ($A\hat{b}in,layer\_p,i(t+\Delta t)$) is inserted here? The use of A and B for (same?) concentrations is highly confusing here.*

**Our reply:** B correspond to a temporary estimation of concentrations and not the final estimation given by Eq. 12. Hence the separation between A and B. A Was replaced by B to avoid the confusion in Eq. 11.

225      *l. 417 - What does "without diffusion" mean here?*

**Our reply:** "but with only one layer and without diffusion" is replaced by "but with only one layer and without representing diffusion inside the aqueous phase"

     *Equation 20 and l. 451: R_diff should probably be R_diff,c (as in Eq. 19).*

**Our reply:** R_diff,c is replaced by R_diff in Eq. 19 for homogenization.

230    *l. 458 - What does "default structure" mean here?*

**Our reply:** The explanation of the default structure is added: the default structure was proposed by Couvidat et al. (2012) in the H$^2$O mechanism for primary compounds and model species with an unknown molecular structure. In this representation, the activity coefficient is set to unity, ensuring that the condensation process is not influenced by non-ideality effects.

235    *l. 465 - "However, at RH=70%, the aerosol is inviscid" - Is the viscosity truly zero here in the calculations, which is what inviscid means, or do just mean its low enough to not affect condensation?*

**Our reply:** "is inviscid" is replaced by "can be considered as inviscid". The viscosity is not zero but it does not affect significantly condensation.

240    *Figure 9 - Why are the initial conditions not the same for these simulations? How is the model initialized? Is the POAmP concentration in the gas phase fixed (open system)?*

**Our reply:** The initial conditions are the same for all simulations. The impression that initial conditions differ between the figures are due to the log scale for time. The figures have been revised to print concentrations starting from the first time step and solve this issue.

245

*l. 483 - It is not clear what is meant with "This type of reaction can account for catalysis by water and pH." - Can the rate coefficient of this reaction be expressed in dependence of humidity and pH?*

**Our reply:** Yes. In that case the reaction rate if multiply by the activity of water or pH. The sentence is replaced by "This type of reaction can account for catalysis by water (reaction rate multiplied by the activity of water) and pH (reaction rate multiplied

250    by pH)."

*l. 530 - "The influence of hydratation is assessed in the Platt isoprene test case with ammonia emission (see previous section)." - Where can this be seen?*

**Our reply:** As the influence of hydratation on glyoxal SOA is quite low, hydratation is not illustrated by a specific figure. The

255    sentence was corrected to: "The influence of hydratation on glyoxal absorption was tested by running the Platt isoprene test case including ammonia emission with and without hydratation."

*l. 536 - "water of pH" should likely read "water or pH"*

**Our reply:** Indeed. Corrected.

*Equation 27 - "H$_e ff$" is incorrectly subscripted*

260    **Our reply:** Corrected. H$_e ff$ replaced by H$_{eff}$.

*l. 573 - "the Henry constant of OH is adapted from Sander (2015) leading to OH aqueous concentration in agreement with typical mean value in the atmosphere from Herrmann et al. (2010)." - It's not clear what the importance of OH is in this section.*

**Our reply:** The parameterization of the irreversible condensation of glyoxal and methylglyoxal represents the impact of the

265 reactions of these dicarbonyls with OH in the aqueous phase of the particles, forming highly soluble and low-volatility com-
pounds, increasing the formation of SOA. The term irreversible is used to distinguish this process from the reversible processes
of hydration and isomerization. The concentration of OH in the aqueous phase is therefore required for estimating $k^l$, the first-
order loss rate of the dicarbonyl in the aqueous phase. The concentration of OH in the aqueous phase is estimated from the gas
concentrations and the Henry constant of OH. The purpose of this parameterization, as well as the role of OH (and therefore
270 its Henry constant), are now explained in the text.

  *l. 588 - It is not clear what is meant with "inorganic aerosol are assumed to be metastable".*

**Our reply:** "Inorganic aerosol are assumed to be metastable" replaced by "Inorganic aerosols are assumed to be thermody-
namically metastable (inorganic compounds are always present in the aqueous phase of particles and no solid salts are formed)
with the exception of $CaCO_3$ that is assumed to have a very low solubility."

275 *l. 593 - "as shown in the guide" - Here and in other places: I think this needs better referencing. What document is this,
where can it be found (URL). The reader might benefit from an explicit page / section reference. In general, I don't think it
makes much sense for this paper to describe a calculation, but then not show results.*

**Our reply:** A reference to the guide was added each time it is mentioned.

  *Figure 13 - It is very hard to see and distinguish all different lines. The legend shows four different experiments, but only 3*
280 *are visible in the figure (dotted, dash-dotted, solid). From the legend, it is not clear which simulation the dash-dotted line refers
to.*

**Our reply:** The figure has been revised and separated into two subfigures (see Fig. 3)

[Figure]

**Figure 3.** Time evolution of $CaCO_3$ particles in presence or in absence of $HNO_3$ with an equilibrium approach (a) or a dynamic approach
(b).

*l. 615 - "An algorithm is used to redistribute the bulk concentrations between the size sections, taking into account the Kelvin effect." - This is not clear, please explain.*

285 **Our reply:** It is replaced by "An algorithm is applied to redistribute the bulk aerosol concentrations across the size sections in a thermodynamically consistent way, explicitly accounting for the Kelvin effect, which increases the effective vapor pressure over smaller particles."

*Section 5.1 - From this section, it does not become clear enough how nucleation works in the model. - Does nucleation*
290 *always occur into the smallest size bin?*

**Our reply:** Yes, nucleation is assumed to occur in the first size bin, which should start around 1 nm. The following sentence is added to the paper: "Nucleated particles are assigned to the first size bin, whose lower bound should be set to about 1 nm to represent freshly formed clusters."

295 *- How is composition of the nucleation size bin treated when nucleation occurs?*

**Our reply:** The composition of the nucleation-size bin is determined by the species involved in nucleation, and it is estimated based on the properties of these species.

*- Can you show / visualize the evolution of the size distribution somehow?*

300 **Our reply:** The evolution of the size distribution for the test that has been added to the paper is available in the paper of SSH-aerosol v1.1 (Fig. 12) and in the guide. Its shows the time evolution of the number distribution for the different size sections.

*- Is nucleation happening in the calculations for Fig. 15?*

**Our reply:** No nucleation was not switched on in these calculations. The words "(nucleation and coagulation are not consid-
305 ered)" are added in the paper.

References

Pöschl, U., Rudich, Y., and Ammann, M.: Kinetic model framework for aerosol and cloud surface chemistry and gas-particle interactions – Part 1: General equations, parameters, and terminology, Atmos. Chem. Phys., 7, 5989–6023, https://doi.org/10.5194/acp-7-5989-2007, 2007.

**References**

Couvidat, F., Debry, E., Sartelet, K., and Seigneur, C.: A hydrophilic/hydrophobic organic ($H^2O$) aerosol model: Development, evaluation and sensitivity analysis, J. Geophys. Res.: Atmospheres, 117, https://doi.org/10.1029/2011JD017214, 2012.

Debry, É., Fahey, K., Sartelet, K., Sportisse, B., and Tombette, M.: Technical Note: A new SIze REsolved Aerosol Model (SIREAM), Atmos. Chem. Phys., 7, 1537–1547, https://doi.org/10.5194/acp-7-1537-2007, 2007.

Iyer, S., Kumar, A., Savolainen, A., Barua, S., Daub, C., Pichelstorfer, L., Roldin, P., Garmash, O., Seal, P., Kurtén, T., and Rissanen, M.: Molecular rearrangement of bicyclic peroxy radicals is a key route to aerosol from aromatics, Nature Comm., 14, 4984, https://doi.org/10.1038/s41467-023-40675-2, 2023.

Sartelet, K., Hayami, H., Albriet, B., and Sportisse, B.: Development and preliminary validation of a modal aerosol model for tropospheric chemistry: MAM, Aer. Sci. and Technol., 40, 118–127, https://doi.org/10.1080/02786820500485948, 2006.

Seigneur, C., Hudischewskyj, A. B., Seinfeld, J. H., Whitby, K. T., Whitby, E. R., Brock, J. R., and Barnes, H. M.: Simulation of aerosol dynamics: A comparative review of mathematical models, Aer. Sci. and Technol., 5, 205–222, https://doi.org/10.1080/02786828608959088, 1986.

---

## Author Comment (AC2)

**Advanced modeling of gas chemistry and aerosol dynamics with SSH-aerosol v2.0**

Karine Sartelet[1], Zhizhao Wang[1,2,3], Youngseob Kim[1], Victor Lannuque[2], and Florian Couvidat[2]

[1]CEREA, Ecole nationale des ponts et chaussées, EDF R & D, Institut Polytechnique de Paris, IPSL, 77455 Marne-la-vallée, France
[2]INERIS, 60550 Verneuil en Halatte, France
[3]Now at University of California, Riverside, CA, USA, and National Center for Atmospheric Research, CO, USA

**Correspondence:** Karine Sartelet (karine.sartelet@enpc.fr), Florian Couvidat (florian.couvidat@ineris.fr)

**Reply to reviewer 2**

*This manuscript describes the development and updates of the model SSH-aerosol, which can be utilized to simulate the formation of secondary aerosols, and evolution of both the primary and secondary aerosols. The manuscript is well written, with the supplement/guide serving as a detailed blueprint for potential users to test an implement the model. The SSH-aerosol model serves as an interesting tool to simulate aerosol dynamics under different environmental scenarios, and would surely benefit the modelling and experimental community. I recommend the publication of this manuscript.*

*This manuscript describes the development and updates of the model SSH-aerosol, which can be utilized to simulate the formation of secondary aerosols, and evolution of both the primary and secondary aerosols. This involves simulating the gas- and particle phase heterogenous chemistry and aerosol dynamics. Various test cases ranging from aerosol dynamic processes and gas-particle phase chemistry were simulated and presented in the manuscript to show model capabilities. The manuscript is well written, with the supplement/guide serving as a detailed blueprint for potential users to test an implement the model. The SSH-aerosol model serves as an interesting tool to simulate aerosol dynamics under different environmental scenarios and would surely benefit the modelling and experimental community. I recommend the publication of this manuscript, after the authors have addressed the following comments/suggestions.*

**General comments:**

*I would recommend adding a figure outlining the different inputs and modules and their workflow. This could also serve the purpose of being a visual reference for any user trying the model.*

**Our reply:** A figure describing the workflow of the model has been added in the section about the model structure. Furthermore, a list of the different model input parameters has been added to the Appendix.

*Add figure letters (e.g. Figure4, a, b or c) to avoid confusion.*

**Our reply:** Figure letters have been added.

25

*While discussing results (e.g. L 303-306, Figure 8 etc.), the authors states what the results shows, but don't discuss why that's happening. The readers will appreciate if they get to know for e.g., why SOA yields are increasing/decreasing in the presence of NOx.*

**Our reply:**

30

*Nucleation: This section has not been discussed in depth but rather just glossed over.*

**Our reply:** To strengthen the section on ultrafine particles, more details about the treatment of coagulation have been added at the beginning of section 5.

"For coagulation, the discretized equations governing aerosol dynamics involve partition coefficients that account for the pos-

35 sibility that the coagulation of particles from two given size sections may produce particle sizes spanning multiple sections. The size mesh is assumed to remain fixed over time, so that these partition coefficients can be pre-computed. In previous model version, the coagulation partition coefficients were calculated using a Monte-Carlo approach. ..... Because the explicit formulation has negligible computational cost compared with the Monte Carlo approach; therefore, only the explicit formulation is retained for internally-mixed particles in v2.0."

40 More details on the size distribution are also added:

"Given the strong impact of coagulation on the growth of ultrafine particles, it should always be included in their modelling. Consequently, the particle-diameter limits are held constant over the course of the simulations. Because nucleation corresponds to the formation of nanometric particles from gas precursors, the lowest particle diameter bound of the size distribution should be about 1 nm."

45 In the nucleation section, the following explanations are also added: "The nucleation parametrizations differ in the gas precursors (sulfuric acid, ammonia, extremely-low volatile organic compounds) and/or in the formulation of the parameterization. ", as well as the formula for organic nucleation:

$$J = scal_{org} \left[ Organics \right]^{nexp_{org}}. \tag{1}$$

Nucleated particles are assigned to the first size bin, whose lower bound should be set to about 1 nm to represent freshly formed

50 clusters. The composition of the nucleation-size bin is determined by the precursor species involved in nucleation, and it is estimated based on the properties of these precursors."

A section describing the growth of ultrafine particles has also been added. "To illustrate the growth of nucleated nanoparticles, the nucleation test case presented in Sartelet et al. (2006) is simulated. The initial distribution corresponds to the hazy

55  conditions of Seigneur et al. (1986). The sulfuric acid production rate is 0.825 $\mu g$ m$^{-3}$ h$^{-1}$, the temperature is 288.15 K and the relative humidity is 60%. The particles are initially assumed to be made of 70% sulfate and 30% ammonium. The initial gas phase ammonia concentration is taken to be 8 $\mu g$ m$^{-3}$. The concentrations of gas-phase ammonia and particulate-phase ammonium evolve with time due to both condensation/evaporation and ternary nucleation. The simulation is run for 1 h with output every 60 s. Fig. 1 shows the size distribution of particles at the initial time and after 1 h. The nucleated particles grow

60  under the effect of condensation of sulfuric acid and ammonia and coagulation. This growth can be accelerated by the presence of extremely low-volatility organic compounds (ELVOCs), such as the SSH-aerosol surrogate "Monomer", underlying the importance of an accurate modelling of the formation of ELVOCs for the growth of ultra-fine particles. "

[Figure]

**Figure 1.** Size distribution of particles in the nucleation test case. Under hazy conditions, nanoparticles nucleated from water, sulfuric acid and ammonia grow to larger size. A simulation adding ELVOCs (Monomer) in the initial conditions is performed to illustrate their effect on the growth of ultrafine particles. Number (left panel, a) and volume (right panel, b) concentrations.

*Model computational time: How much impact does different configurations (gas+SOA schemes) have on the computational time. Since this model can be couple to a 3-d model, how realistic is it to use particle phase and viscosity calculation with the*

65  *coupled 3-d model since its includes discretised particle layer calculations.*

**Our reply:** A table on the model computational time has been added and discussed. While being very time consuming, accounting for diffusion inside layers is doable as a first study on considering viscosity into 3D was performed by Kim et al. (2019).

70 **Minor comments:**

*L 35: "atmospheric chambers"- I believe the correct terminology would be atmospheric smog chambers.*
**Our reply:** Yes, replaced

*L 43: "Several aerosol box models exist". I think this line is unnecessary. Either remove it or club it with the next line.*
75 **Our reply:** The sentence has been replaced by "Several aerosol box models exist have been developed, and most represent aerosol dynamics using a sectional approach..."

*L56-57: One must be careful in formulating such descriptions. MCM is an explicit gas phase oxidation mechanism, not a SOA formation scheme. It is coupled to a SOA formation scheme and therefore should be categorized as a chemical scheme*
80 *and not a SOA scheme.*
**Our reply:** The sentence starting by "For the formation of secondary organic aerosols,..." has been replaced by "For the formation of organic oxidation products that could be semi-volatile and partition to form secondary organic aerosols (SOAs), ...".

*L65-66: Some examples of different reduction strategies could be included.*
85 **Our reply:** Examples of reduction strategies have been added (lumping, replacing, jumping, removing).

*L89: "repertories" to repositories*
**Our reply:** "repertories" has been replaced by "directories"

90 *L 99: what kind of solver is used to deal with the ODE systems? How will it affect the computational time?*
**Our reply:** A subsection has been added in section 2 to detail the numerical implementation. A reference to this subsection is added here.

*L 110: How are the saturation vapor pressures generated? Nannolal/Evaporation/SIMPOL or another mechanism?*
95 **Our reply:** In SSH-aerosol v2.0, saturation vapor pressures are input data, they can be generated by various methods, such as those described by the reviewer. For clarity, the following sentence is added to the text: "The saturation vapor pressures are provided as model inputs and may be derived through different methods or parameterizations, including tools such as UMan-SysProp (Topping et al., 2016)."

100 *L211 – 213: does this mean that one can use 2 or more chemistry schemes at the same time? For example, one can use CB05 for estimating a reaction products conc., say X and another explicit scheme to use the [X] in their respective chemical schemes? If so, does it only work for first generation oxidation products or more complex oxidation products as well? Are these conc. profiles interpolated to the timestep of the SOA chemical scheme? L214-216: This part is unclear. Does it mean*

*one can have a specific combination for each precursor + SOA scheme? E.g. alpha-pinene using MCM + beta caryophyllene using RACM2? if so how would one account for example RO2 from ap +RO2 (bcarp) reaction products from different chemical schemes, in case the resulting RO2 is not present in either scheme?*

**Our reply:** For clarity this paragraph was rewritten: "Depending on the user's settings, oxidant concentrations can either be prescribed using constant input profiles, or they can be computed using existing gas-phase chemistry mechanism, such as CB05, RACM2 or MELCHIOR2. These schemes can be complemented with SOA chemical schemes describing the degradation of VOCs leading to SOA formation. To offer flexibility, the model allows users to construct customised chemical mechanisms, covering both gas-phase and SOA chemistry, by combining a chosen gas-phase mechanism with one or more SOA schemes (typically one per SOA precursor class). For VOCs already represented explicitly in the selected gas-phase mechanisms (e.g. toluene, monoterpenes), the associated SOA scheme can be added without modifying oxidant concentrations. However, some VOCs are not included in these gas-phase mechanisms, in which case the inclusion of specific reactions for radicals, oxidants as well as SOA formation may be necessary. For example, the impact of naphthalene on ozone and radical production is not represented in the CB05, RACM2 or MELCHIOR2 mechanisms."

*L223-224: There are different vapor pressure estimation methods on the UManSysProp. Which method was used?*

**Our reply:** The following was added "often using the vapor pressure estimation methods ('v0' (Myrdal and Yalkowsky, 1997)), with the boiling point estimation methods('b0' (Nannoolal et al., 2004))".

*L234: is this analogous to "lumped RO2" species? Is this RO2 pool rereferring to RO2 from one precursor alone or can it refer to lumped RO2 from all of MCM or a subset of precursors?*

**Our reply:** "that include $RO_2$ from all precursors" is added after $RO_2$ pool.

*L 236: what does "ARR" stand for?*

**Our reply:** ARR indicates it is an Arrhenius type of reaction. This has been added to the text.

*L 243: "TBRO2" is not referenced in the equation above or explained anywhere else.*

**Our reply:** This sentence has been removed.

*Figure 2: This makes sense since MCM doesn't yet include a fully developed peroxy radical autooxidation scheme for beta caryophylle. It would be interesting to perform such tests with apha-pinene.*

**Our reply:** We agree with the reviewer that $\alpha$-pinene is a valuable reference system for exploring $RO_2$ chemistry, particularly under realistic atmospheric conditions where multiple precursors contribute.

*Section 3.2 My one suggestion here to improve readability would be to have a table detailing the explicit and reduced mechanism, with a shorthand representing each. For e.g near explicit mech for monoterpenes could be denoted by Expmcm+pram or something like that. It could make it easier for readability without one having to go back and forth to see what the respective*

| Precursor | Mechanism name | Type | Reference |
|---|---|---|---|
| Naphthalene | $H^2O$ | Implicit | Couvidat et al. (2013) |
| | Expl. | Quasi-explicit | Lannuque and Sartelet (2024) |
| Toluene | $H^2O$ | Implicit | Couvidat et al. (2012) |
| | Expl. | Quasi-explicit with | Lannuque et al. (2023) |
| | | ipso-BPR molecular rearrangement | + Sartelet et al. (2024) |
| | Rdc | GENOA Reduced | Sartelet et al. (2024) |
| Monoterpenes | $H^2O$ | Implicit | Couvidat et al. (2012) |
| | Expl. | Quasi-explicit (MCM + PRAM) | Roldin et al. (2019); Wang et al. (2023) |
| | Rdc | GENOA Reduced | Wang et al. (2023) |
| $\beta$-caryophyllene | $H^2O$ | Implicit | Couvidat et al. (2012) |
| | Expl. | Quasi-explicit (MCM) | - |

**Table 1.** SOA chemical mechanisms used in the test cases of section 3.2

*mechanisms were.*

**Our reply:**

The following table (Table 1) has been added to section 3.2:

*L281: This should be right panel.*

**Our reply:** Yes, thank you.

*L 294: upper right panel?*

**Our reply:** Yes, modified.

*L296-298: Referring to Fig 4: these differences are barely visible in the plot. Perhaps the authors will think about providing percentage increase or decrease w.r.t the reference to indicate the difference.*

**Our reply:** The sentence "They are also lower than the reference with higher NO$_2$ levels in the first 1.5 hours of the simulation but higher after that, in opposition to the simulation with the near-explicit scheme." corresponds to small variations that are very meaningful and hence it has been removed.

*L 299-303: why is that?*

**Our reply:** For naphthalene, the difference between the concentrations simulated with the near-explicit and the $H^2O$ scheme reflect large uncertainties in the $H^2O$ scheme, which was built by (Couvidat et al., 2013) from a specific set of experiments from (Chan et al., 2009) that are not representative of the full range of atmospheric conditions. Even though these experiments were corrected from wall-loss effects of particles, they were not corrected for gas wall losses. Furthermore, the behaviour of the $H^2O$ scheme may not fully extrapolate to all experimental setups, especially flow tubes where residence times are much

160    shorter than in smog chambers. This was added in the text.

*Figure 3: what is the reference scheme? it should be mentioned in the caption. Is the related to NO2 conc.?*

**Our reply:** The following has been added in the caption: (Ref. denotes the reference $NO_2$ levels, $NO_2$ x 2 the simulation with doubled $NO_2$, and $NO_2$ / 2 the simulation with halved $NO_2$).

165

*L 364-365: Since H2O is based on smog chamber experiments, was wall losses considered when simulating SOA for species which was compared to H2O? I suspect the SOA yields for species where H2O was used for comparison would differ if wall losses of organics would be considered.*

**Our reply:** $H^2O$ was built using smog chamber experiments after correction of wall losses for particles, but generally these

170    corrections do not account for wall losses of gases. Wall losses for gases and particles are considered in the simulations performed in the current study, independently of the scheme used.

*L 378-380: why does the H2O mechanism overestimate the SOA mass?*

**Our reply:** Each precursor-specific $H^2O$ mechanism was developed using a limited number of chamber experiments performed

175    under particular conditions. Its behaviour may therefore not fully extrapolate to all experimental setups, especially flow tubes where residence times are much shorter than in smog chambers. This difference in conditions may contribute to the apparent overestimation of SOA mass.

*L398: "interface the index of the layer at the interface". what does this represent or mean? Also K,bin, interface, p,i and the*

180    *corresponding Mo bin,interface layer is not defined.*

**Our reply:** Quotes were missing. The phrase should be read like "interface" the index of the layer at the gas/particle interface. Layer has to be replaced by interface for the values of variables at the interface

*L 413: "f". This needs more explanation here. For e.g., what does a value f=0.2 indicate? how is f determined?*

185    **Our reply:** f is a weighting factor between two iterations. The algorithm is presented into greater details in Couvidat and Sartelet (2015). Please refer to Couvidat and Sartelet (2015).

*L 430: "Compounds not affected to a specific molecular structure". what does this mean?*

**Our reply:** The sentence was deleted.

190

*Figure 9: I assume that since this is an organic phase there would be no charged compounds would be present in the particle phase. But when the RH increases compounds in the particle phase are bound to break and form cations and anions. How will this affect the viscosity especially in the presence of acids?*

**Our reply:** Dissociation of acid in the organic phase is currently not taken. The implementation of this process will be consid-

 ered for future version.

*Figure 9: Add panel letters to make it easy to follow. how does the c panel which is bottom left change if you have varying RH for alcohol+acid + varying RH?*

**Our reply:** Panel letters added. As there are no particular treatment for acids on viscosity, the effect of humidity will be similar to panel b. The increase of RH will lead to a faster condensation.

*L480: parameter k. Is this a function of water conc. and pH?*

**Our reply:** The text now states: "This type of reaction can account for catalysis by water (reaction rate multiplied by the activity of water) and pH (reaction rate multiplied by pH)."

*L 505: Kmaxoligo , hoe is this determined?*

**Our reply:** As mentionned later, all parameter for the bulk oligomerization parameterization are coming from Couvidat et al. (2018a). In this study, the parameters were determined by running a 0D model and were fitted in order to reproduce the evolution of observed molar masses of oligomers during an experiment.

*L505: amonomer, I would suggest representing this as Eta or sum(aA, monomer), both in the equation 24 and here.*

**Our reply:** $a_{monomer}$ was replaced by $\sum_i a_{i,monomer}$.

*Eq. 28: is this nomenclature based on H2O or MCM or other chemical schemes?*

**Our reply:** This nomenclature was specifically chosen for SSH-aerosol.

*L 587: SOAP. Does the model account for the formation of salts in the particle-phase?*

**Our reply:** The text now states: "Inorganic aerosols are assumed to be thermodynamically metastable (inorganic compounds are always present in the aqueous phase of particles and no solid salts are formed) with the exception of $CaCO_3$ that is assumed to have a very low solubility."

*Figure 13: please improve the legend. It's hard to read it in its current form.*

**Our reply:** The figure has been separated into two subfigures (see Fig. **??** in the response to comments from Reviewer 1).

*Nucleation: Have the authors considered to couple ACDC for more complex termolecular acids and bases?*

**Our reply:** Yes that would be interesting for a better representation of nucleation, and we will consider it in further version of the model.

**References**

[revised manuscript text omitted]

---

## Referee Report (RR1)

**Review**

1. The following question has not been addressed
   While discussing results (e.g. L 303-306, Figure 8 etc.), the authors states what the results shows, but don't discuss why that's happening. The readers will appreciate if they get to know for e.g., why SOA yields are increasing/decreasing in the presence of NOx.

2. Although I appreciate the use of ug m-3 for gas phase concentrations, it would intuitively make sense to address the gas-phase concentrations of species in ppm or ppb.

3. L296-298: *Referring to Fig 4: these differences are barely visible in the plot. Perhaps the authors will think about providing percentage increase or decrease w.r.t the reference to indicate the difference.* Our reply: The sentence "They are also lower than the reference with higher NO2 levels in the first 1.5 hours of the simulation150 but higher after that, in opposition to the simulation with the near-explicit scheme." corresponds to small variations that are very meaningful and hence it has been removed.

   Why was it removed if was meanigful?

---

## Author Response (AR2)

**Advanced modeling of gas chemistry and aerosol dynamics with SSH-aerosol v2.0**

Karine Sartelet[1], Zhizhao Wang[1,2,3], Youngseob Kim[1], Victor Lannuque[2], and Florian Couvidat[2]

[1]CEREA, Ecole nationale des ponts et chaussées, EDF R & D, Institut Polytechnique de Paris, IPSL, 77455 Marne-la-vallée, France
[2]INERIS, 60550 Verneuil en Halatte, France
[3]Now at University of California, Riverside, CA, USA, and National Center for Atmospheric Research, CO, USA

**Correspondence:** Karine Sartelet (karine.sartelet@enpc.fr), Florian Couvidat (florian.couvidat@ineris.fr)

**Reply to reviewer 1**

*The authors did a good job in addressing my reviewer comments and I can recommend publication of the manuscript. Here are a few minor points where communication remains unclear in my opinion:*

*l. 462: "The main consequence of this simplification is that the diffusion of a compound is calculated as if the compound has the same affinity with every layers." - This is unclear, what is meant with "affinity" here?*

**Our reply**: affinity refers to the activity between the compound, which diffusion is calculated, and the compounds of the particle phase. The sentence is replaced by "The main consequence of this simplification is that the diffusion of a compound is computed assuming identical affinity across all layers, where this affinity represents the compound's activity relative to the other particle-phase constituents."

*l. 609: "reaction rate multiplied by pH" - Do you mean this literally, as in a simple product? Is the reaction rate so directly correlated with pH?*

**Our reply**: "reaction rate multiplied by pH" is replaced by "reaction rate multiplied by the activity of $H^+$ ion"

*Figure 10: Is there also exchange between the intermediate layers and the interface layer? I cannot see an arrow between them.*

**Our reply**: One of the arrows was placed between the core layer and the interface layers instead of being placed between the intermediate layers and the interface. As the implicit method of Couvidat and Sartelet (2015) does not represent explicitly diffusion, The process is represented by a exchange fluxes between the interface layer and inner layers (that are represented by the double-headed arrows at the bottom of the figure).

The new version of the figure:

[Figure]

**Figure 1.** Illustration of implicit treatment of diffusion inside particle.

**Reply to reviewer 2**

*1. The following question has not been addressed: While discussing results (e.g. L 303-306, Figure 8 etc.), the authors states what the results shows, but don't discuss why that's happening. The readers will appreciate if they get to know for e.g., why SOA yields are increasing/decreasing in the presence of NOx.*

**Our reply**: The following sentence is added line 331: "Differences in the simulated concentrations and their temporal evolution among the implicit, quasi-explicit and GENOA-reduced schemes arise from the richer formation pathways included in the quasi-explicit and GENOA-reduced mechanims." The following sentence is added line 346: "The higher concentrations may be attributed to $RO_2$-$HO_2$ reaction rates, which tend to yield more highly oxidized monoterpene degradation products."

*2. Although I appreciate the use of ug m-3 for gas phase concentrations, it would intuitively make sense to address the gas-phase concentrations of species in ppm or ppb.*

**Our reply**: We deliberately use $\mu g\ m^{-3}$ for all species, including gas-phase compounds, in order to maintain consistency across particulate and gaseous pollutants. Expressing concentrations in mass units is standard practice in atmospheric chemistry and air-quality modelling. It avoids ambiguities related to temperature and pressure assumptions inherent to volume-based units (ppm or ppb), ensuring internal consistency across the modelling framework. For these reasons, we retain $\mu g\ m^{-3}$ throughout the manuscript. It is also the main concentration unit used within the code.

*3. L296-298: Referring to Fig 4: these differences are barely visible in the plot. Perhaps the authors will think about providing percentage increase or decrease w.r.t the reference to indicate the difference. Our reply: The sentence "They are also lower than the reference with higher NO2 levels in the first 1.5 hours of the simulation but higher after that, in opposition to the simulation with the near-explicit scheme." corresponds to small variations that are very meaningful and hence it has been removed. Why*

*was it removed if was meanigful?*

**Our reply**: Yes, indeed, the differences are barely visible in the plot and we consider them not to be significant. We also note that in our previous reply, the word "not" was inadvertently omitted. The sentence "They are also lower than the reference with higher $NO_2$ levels in the first 1.5 hours of the simulation but higher after that, in opposition to the simulation with the near-explicit scheme" referred only to very small variations that are **not** meaningful from a scientific perspective. For clarity and to avoid over-interpretation, this sentence has therefore been removed from the revised manuscript.